# Evolving public behavior and attitudes towards COVID-19 and face masks in Taiwan: A social media study

**Chih-Yu Chin**[1]*, **Chang-Pan Liu**[2,3], **Cheng-Lung Wang**[4]

**1** Department of Information Management, Chung Yuan Christian University, Taoyuan, Taiwan,
**2** Department of Internal Medicine, MacKay Memorial Hospital, Taipei, Taiwan, **3** Department of Medicine, MacKay Medical College, New Taipei, Taiwan, **4** Big Data Co., Ltd., Taipei, Taiwan

* king@cycu.edu.tw

## Abstract

Facing the COVID-19 pandemic, Taiwan demonstrated resilience at the initial stage of epidemic prevention, and effectively slowed down its spread. This study aims to document public epidemic awareness of COVID-19 in Taiwan through collecting social media- and Internet-based data, and provide valuable experience of Taiwan's response to COVID-19, involving citizens, news media, and the government, to aid the public in overcoming COVID-19, or infectious diseases that may emerge in the future. The volume of Google searches related to COVID-19 and face masks was regarded as an indicator of public epidemic awareness in the study. A time-series analysis was used to explore the relationships among public epidemic awareness and other COVID-19 relevant variables, which were collected based on big data analysis. Additionally, the content analysis was adopted to analyze the transmission of different types of fear information related to COVID-19 and their effects on the public. Our results indicate that public epidemic awareness was significantly correlated with the number of confirmed cases in Taiwan and the number of news reports on COVID-19 (correlation coefficient: .33–.56). Additionally, the findings from the content analysis suggested that the fear of the loss of control best explains why panic behavior occurs among the public. When confronting the highly infectious COVID-19, public epidemic awareness is vital. While fear is an inevitable result when an emerging infectious disease occurs, the government can convert resistance into assistance by understanding why fear arises and which fear factors cause excessive public panic. Moreover, in the digitalization era, online and social media activities could reflect public epidemic awareness that can e harnessed for epidemic control.

## Introduction

The coronavirus disease 2019 (COVID-19) epidemic broke out in December 2019 in Wuhan, Hubei, China, and then spread around the world. COVID-19 was declared a worldwide pandemic on March 11 [1], and as of the end of March, approximately 750,890 people worldwide

**Data Availability Statement:** All relevant data are within the paper and its Supporting Information files.

**Funding:** This manuscript was supported by the Ministry of Science and Technology, Taiwan, under

grant number MOST 108-2410-H-033-043. The
corresponding author, CY, is the principal
investigator of the research project. The URL of the
funder website is https://www.most.gov.tw/?l=en.
Finally, the funder had no role in study design, data
collection, and analysis, decision to publish, or
preparation of the manuscript. On the other hand,
the funder of Big Data Co., Ltd. provided financial
support in the form of salaries for the author, CL,
and played a role in data collection in the study.
The funder provided a social media search engine
research tool to allow authors to collect data from
social media. The URL of the funder website is
http://bigdata.com.tw/en/.

**Competing interests:** The funder of Big Data Co.,
Ltd. provided financial support in the form of
salaries for the author, CL. It offered a part of the
study's research materials. The support does not
alter our adherence to PLOS ONE policies on
sharing data and materials.

were infected, and more than 170 countries had been affected [2]. COVID-19 has become the
world's leading threat; therefore, implementing effective strategies to prevent its spread has
become a topic of discussion and research by governments and academia worldwide. Owing
to its geographical location and historical background, Taiwan is closely associated with China
[3]. Taiwan was expected to be the second-highest risk for COVID-19 outside China [4]. As of
March 31, there were 322 confirmed cases of COVID-19 in Taiwan. Of the confirmed cases,
276 were imported, and 46 were indigenous [5]. Regarding the total confirmed cases of
COVID-19, Taiwan ranked 79th worldwide [6], suggesting that the Taiwanese government
achieved early control of both imported and indigenous cases. Previous literature has attrib-
uted the successful early control to the Taiwan government's quick responses and countermea-
sures to COVID-19 such as border control from the air and sea, case identification (using new
data and technology), and quarantine of suspicious cases [3]. However, COVID-19 is very con-
tagious and is infectious even through asymptomatic carriers [7]. Thus, apart from govern-
ment measures, we argued that earlier public epidemic awareness in Taiwan may be another
key to comprehensive prevention. The study aims to fill in the story of Taiwan's experience of
defending against COVID-19 at an early stage from the perspectives of public epidemic aware-
ness measured by disease-related information-seeking by the public and the prominent role of
media in the transmission of COVID-19-related information. Also, the study documents the
government's face-masks countermeasures against COVID-19 and how it associates with pub-
lic epidemic awareness. This might be instructive for other countries in dealing with the next
crisis resulting from emerging infectious diseases.

Public epidemic awareness has been recognized as a crucial determinant of public epidemic
prevention [8–10]. When people are aware of the importance of avoiding the epidemic or are
concerned about getting infectious, they will protect themselves with preventive behaviors
such as handwashing, using hand disinfectant, avoiding contact with their face, gargling,
attending to healthcare, and collecting flu-related information [8, 9]. Recent research on public
health awareness during the COVID-19 epidemic also showed that one's awareness of how to
prevent COVID-19 significantly contributes to one's behavioral chance of fighting against
COVID-19 [e.g., 10]. Given the importance of public epidemic awareness for epidemic con-
trol, leaders and governments must understand how to leverage public epidemic awareness to
combat COVID-19.

During a public health crisis, such as emerging infectious disease, effective health commu-
nication is vital for the rise of public epidemic awareness, where government and media play
crucial roles [11–14]. As for a government, it is essential to adopt an effective communication
strategy with helpful content, trustworthy sources, and efficient channels to enable the public
to be aware of or learn about disease-related information promptly. Previous research has indi-
cated the higher public's exposure to government information about COVID-19, the public
would have higher probabilities to take necessary preventive measures to protect themselves
from being infected by COVID-19 [11]. On the other hand, drawing from the literature on
agenda setting, mass media plays a prominent role in society in that what people perceive as
being important to society is affected by the media agenda [15]. This is also the case when it
comes to infectious disease. Mass media are an important channel for people to receive dis-
ease-related information, and also serve as one of the important strategies for governments or
public health agencies to communicate preventive measures to the public [12]. Previous
research used a mathematical model of disease transmission to verify that mass media had a
positive impact on the spread of the H1N1 epidemic; more media coverage may have led to a
reduced peak size and the final degree of influence of the epidemic [13]. Research has also
found that the sooner and the louder mass media report the information of infectious disease
at an outbreak, the better the citizens can prepare to prevent infection [14].

Additionally, due to the rapid development of mobile Internet technology, the way that people receive the disease-related information during the epidemic of COVID-19 is to some extent different from the scenario of previous epidemics, such as the severe acute respiratory syndrome (SARS) epidemic in 2003 and the H1N1 pandemic in 2009. Nowadays, disease-related information will quickly diffuse via online social networks when an infectious disease emerges and spreads within a population. People who receive the COVID-19 information could play an active role in sharing the information via various social media platforms such as Facebook, Instagram, and Twitter, and these social media platforms have a significant positive influence on public epidemic awareness and public behaviors of epidemic prevention [10].

In addition to the quantity of information diffusion, the way COVID-19 is portrayed in the media and to the public determines the rise of public epidemic awareness and public health behavior changes. It means that the content of the message transmitted to people also has a great impact on how people think about and respond to COVID-19. In the literature of public health campaigns and risk perception, the fear-embedded message is more powerful to per-suade people to take actions avoiding hazards than that without embedding fears [16]. The mechanism is that fear could arouse vigilance against threats to an individual, and this leads the individual to employ measures to protect him/herself from harm [16]. Many public health information campaigns often use fear to improve public health management [17]. Previous studies have discussed the role of media in propagating the fear of COVID-19 via infodemic and sensational news reports [18, 19]. However, little is known about which types of fear-embedded messages serve the function of stimulating the public perception of risk and enhancing public epidemic awareness.

Understanding what types of messages are associated with various characteristics of fear spread to people and play a role in public risk perception during an epidemic is helpful for the government to enact appropriate strategies to arouse citizen's fear or mitigate panic. "The same knife cuts bread and fingers," is truly applicable in the case of the epidemic of fear. While the public's fear arousal is somewhat beneficial for epidemic control, excessive anxiety and fear in society in the face of emerging infectious diseases or major disasters may lead to more social, psychological, and economic problems, such as a decrease in the willingness to engage in human-to-human interactions [20], prejudice and stigma [18, 21], and panic buying [22]. Effective control over fear of the COVID-19 epidemic to improve the public's infection control awareness while avoiding excessive panic, which may result in external effects on the society and economy, is an important matter that the government must consider during epidemic control [23].

With the importance of public epidemic awareness of emerging infectious diseases, how it is monitored is particularly important for a government to dynamically adjust its response to the public. Previously, cross-sectional telephone surveys were used to measure public aware-ness of diseases [24, 25]. While it provided valuable information about public perceptions at the time, it is hard to survey continuously throughout the spread of the epidemic due to cost-effectiveness issues. With the development of the Internet and information communication technology, a web-based methodology has been used to promptly capture public awareness. Browsing, commenting, and various online behaviors represent free thoughts, so the user's online activities can be tracked and recorded. Different online actions are evaluated to explore public awareness of diseases, thereby providing new perspectives to monitor public health [26]. Researchers have used the methodology to observe correlations between public anxiety and flu-related information-seeking behavior [27] or between the number of confirmed influ-enza cases and disease information-seeking behavior to examine whether influenza outbreaks can be detected early through disease information-seeking behavior [28]. These early studies used to view the disease-related information-seeking behavior as a tool for monitoring an

outbreak of disease. However, recent research started to regard searching behaviors as a demonstration of public awareness [29]. Previous research on using searching behaviors to capture public epidemic awareness mainly focused on the disease itself. Few studies have investigated the searching behavior of personal protective equipment and how it could play an important role in epidemic control as another essential indicator of public epidemic awareness. Thus, to detect the social response to emerging infectious diseases, we argued that these online behaviors (i.e., searching for disease and personal protective equipment) not only capture public opinion in a timely manner, but also reveal changes in public attention and attitudes to disease-related issues. The collection and interpretation of such information are helpful to governments managing public crises.

Combining the above discussion, this naturalistic study aimed to document real-time public epidemic awareness of COVID-19 through collecting disease-related information-seeking behaviors on Google in Taiwan and examines its dynamic relations to epidemic development, disease-related information spread on the Internet, and government face-mask related policies during the initial phase of the spread of COVID-19 in Taiwan. Thus, we reviewed how public epidemic awareness changed in tandem with the rising number of confirmed cases of COVID-19 worldwide and in Taiwan, the number of news reports on COVID-19, and the volume of mentions of COVID-19 and face masks on social media between December 31, 2019, and February 29, 2020. Additionally, we discuss how these variables were associated with government face mask-related policies in Taiwan. Moreover, to fill the gap in the literature on the diffusion of various fear-related messages on the Internet and their effects on public epidemic awareness along with the development of the COVID-19 epidemic. We conducted a content analysis to investigate the transmission of five types of fear information of COVID-19, including mistrust, severity, loss of control, uncertainty, and susceptibility, drawing from previous literature on fear and risk perception [16, 23, 30–32], during the early stage of the spread of COVID-19 from December 31, 2019, to March 29, 2020, and examine how they are associated with the public epidemic awareness. By doing so, the study obtained valuable experience based on Taiwan's response to COVID-19, involving citizens, news media, and the government, to aid the government or public in overcoming COVID-19, or infectious diseases that may emerge in the future.

To preview the overall findings in the study based on Taiwan's experience against COVID-19 in the early stage, the study concluded that early public epidemic awareness immediately rose in Taiwan since the news media and government attached great importance to the COVID-19, and a web-based methodology could serve as a timely method to capture the dynamic change of public epidemic awareness in the course of epidemic spread. It would be an instrumental guide for government decision making. Moreover, the different types of fear embedded in the news information play different roles in various phases of the spread of COVID-19. The amount of *loss of control* news information was greater than the news information embedded with other types of fears at the stage of high Google search volume for face masks. This gives us an insight into the matter of supply management and crisis communication during the outbreak, since once individuals lose control over their self-protection, it is easier to cause panic buying, which creates greater burdens for society.

## Methods

Since this study aims to provide a timely method to capture public epidemic awareness and discuss how it is associated with the spread of COVID-19, news reports on COVID-19, and the government's countermeasures against COVID-19, we adopted the web-based methodology to collect the COVID-19-relevant digital footprint online, as introduced below. The study

consisted of two stages of analysis to address different questions. The first is about the trend of public epidemic awareness and other external variables associated with COVID-19 that we focused on in the study (e.g., the confirmed cases of COVID-19), and the second concerns what types of fear-arousal news information are associated with public epidemic awareness. Since Mandarin Chinese (Traditional Chinse) is the official language in Taiwan, the following word-related data collected in the study were based on Mandarin Chinese.

### The first stage of the study

**Data collection.** The first stage of the study aims to address the issue of how the earlier public epidemic awareness rose and changed in the first stage of the COVID-19 epidemic spread. Thus, we continuously collected data related to the following seven indicators between December 31, 2019, and February 29, 2020: total global number of confirmed cases of COVID-19, total number of confirmed cases of COVID-19 in Taiwan, the number of news reports on COVID-19, the volume of mentions of COVID-19 on social media, the volume of mentions of face masks on social media, Google search volume for COVID-19, and Google search volume for face masks. Among them, the Google search volumes for COVID-19 and face masks were used to measure public epidemic awareness. The disease information searching behaviors have been used as an indicator of public epidemic awareness recently [29]. It reflects the behavioral intentions of individuals. In addition to searching for disease, given that related treatment drugs and vaccines for COVID-19 were under clinical trials in the early stage of COVID-19 [33], face masks have become necessary personal protective equipment for managing infectious diseases [34, 35]. Increasing intended behaviors of searching for face masks indicates that the public is aware of the threat and reflects other positive hygiene practices [36, 37]. Thus, we examine the public epidemic awareness by measuring the public online information-seeking behavior of COVID-19 and face masks.

The total global confirmed cases data were mainly collected through the real-time global cases website of System Science and Engineering Center, Johns Hopkins University [38]. Because the site was not completed until January 20, the number of global confirmed cases between December 31 and January 19 was based on WHO released data (https://covid19.who.int/). The total number of confirmed cases in Taiwan was based on the daily press conference convened by the Taiwan Centers for Disease Control (TCDC) and simultaneous press releases [39].

Data on the number of news reports on COVID-19, the volume of mentions of COVID-19 on social media, and the volume of mentions of face masks on social media were collected through the KEYPO Big Data Analytics Engine [40], Taiwan's well-known online public opinion system. The former represents the number of news reports on COVID-19, whereas the latter two represent the number of mentions of COVID-19 and face masks in the posts and comments on social media. A higher number of mentions on social media indicates a higher degree of discussion. The detailed information about the scope of the data and KEYPO big data analytics engine can be seen in S1 Appendix.

The Google search volumes for COVID-19 and face masks were acquired through the Google Trends search service, with the search area set to Taiwan and the search time set within the research period. Google Trends provides a normalized value according to the set time range, with a scale of 0–100. A daily search volume of 100 is the highest value for the keyword within that time range. Regarding the data processing, Google Trend eliminates repeated searches from the same person over a short period of time. To compare the above indicators in a trend chart with the same scale, other indicators were normalized using the same method, with the scale ranging from 0 to 100.

In addition to Taiwan, the Google search volumes of face masks in infected East and Southeast Asian countries, including Singapore, Malaysia, Thailand, Vietnam, South Korea, and Japan were collected to make a cross-national comparison in the study. These countries shared common conditions that had resisted the COVID-19 epidemic for more than 30 days from the date that the countries identified the first confirmed case to February 29, 2020. Moreover, the total number of confirmed cases of COVID-19 was greater than 15 on February 29, 2020. For a better illustration of cross-national comparison, we rescaled each data point by dividing by the mean of Google search volume for "face masks" from December 31, 2019, to January 9, 2020 (the base).

**Statistical analysis.**    Apart from plotting the trend chart to depict the synchrony among the seven indicators, the study adopted time series analysis (TSA) conducted with SPSS version 22 to deal with the nature of time series data. Specifically, three functions, namely, the autocorrelation function (ACF), the partial autocorrelation function (PACF), and the cross-correlation function (CCF) were implemented. In our study, the time series data might yield temporal dependence, which would produce spurious relationships when conducting simple linear correlations [41]. Given that, the first two functions, ACF and PACF, were used to identify the temporal dynamics of an individual time series variable and help researchers to examine whether the errors of a certain series variable would correlate itself over time. If a certain variable has autocorrelative relationships, the series data was whitened by removing the autocorrelation before conducting the CCF. The CCF was used to examine the dynamic relationship between two time series variables by calculating the correlation coefficients between two time series variables contemporaneously and at various lagged values. It gives us an insight into how two time series variables coincidently related to each other, and whether movement in one variable tends to precede or follow movement in the other. The value of CCF indicates the linear relationship as a number between -1 (negatively correlated) to 0 (not correlated) to 1 (perfectly correlated).

## The second stage of the study

**Data collection.**    To further investigate what types of fear were embedded in the COVID-19-related popular events in Taiwan, and how these types of fear had an impact on public epidemic awareness across the periods of epidemic spread from December 31, 2019 to March 29, 2020, the study adopted a content analysis to identify the COVID-19-related information.

To select representative popular events among the past 90 days, we screened COVID-19-related popular daily events by using the KEYPO Big Data Analytics Engine. The time range for popular event samples was from 31 December 2019 to 29 March 2020, and the total sample size was 359 events.

Categorization is to classify the content of the study subjects into groups in order to endow symbolic meaning. Regarding the categorization of fear, although there are various differences in an individual's ability to determine whether an event is a threat, many researchers who study risk perception and fear believe that people tend to fear similar things due to similar factors based on human instinct [16,30–32]. After reviewing these studies and examining collected events, the study constructed "fear type" categories, including *mistrust*, *severity*, *loss of control*, *uncertainty*, *susceptibility*, and *without fear*. Mistrust hypothesizes that the less we trust the people who are supposed to protect us, the more fear we feel. Regarding the severity and susceptibility, the two elements are frequently used as fear appeals in the previous public health campaign research. When people perceive something that will cause a negative consequence (e.g., cause one's permanent damage to health) or people perceive their own vulnerability to an illness or disease, people will feel fear and try to protect themselves [16]. As for the loss of

control, if people can control the risk of disease (e.g., having personal protective equipment), then they will not regard this disease as a threat. Finally, when we are more uncertain about something that may be harmful, we will be more attentive or fearful of the thing. For details of how events are grouped into each category, please refer to S1 Table.

**Statistical analysis.** In order to examine the associations between different types of fear events and public epidemic awareness (i.e., Google search volume for face masks). One-way ANOVA was used. This was done to analyze whether there were differences in the Google search volume for face masks on a day when different types of fear events occurred. Scheffé's method was used for post hoc multiple comparisons. Since Levene's test for equality of variances was found to be violated for the present analysis, $F(5,353) = 12.14$, $p < .05$, the study used the bootstrap method, a resampling method, to derive parameter estimates of standard errors and confidence intervals, and to correct biases [42].

Additionally, to extend our analysis to the correlation between Google search volume for face masks at different periods and the number of various types of fear events, we used a threshold value of 25 for the Google search volume, which was used to divide the period into three stages. These periods are namely: the first stage (2019/12/31–2020/1/26, 27 days), the second stage (2020/1/27–2/22, 27 days), and the third stage (2020/2/23–3/29, 36 days). There are significant differences in the Google face mask search volumes in these three stages according to the One-way ANOVA (S2 Table). The first stage is the low search volume group ($M = 4.33$, $SD = 6.87$), the second stage is the high search volume group ($M = 45.19$, $SD = 21.47$), and the third stage is the medium search volume group ($M = 25.55$, $SD = 5.20$). After differentiating these three stages, the Chi-square test and multinomial logistic regression were used to examine differences in the number of different types of fear events among the different stages.

# Results

## Imported overseas cases in the early stage: The rise of public epidemic awareness

Table 1 shows the strongest paired cross correlations (with day time lags between -7 and 7) among Taiwanese confirmed cases, the volume of the mentions on social media, the number

**Table 1. The cross-correlation analysis among the six time series variables after removing the first order autocorrelation.**

| | 2[b] | 3[b] | 4[b] | 5[b] | 6[b] |
|---|---|---|---|---|---|
| 1. Total number of confirmed cases of COVID-19 in Taiwan[a] | .34 (lag 0) | .35 (lag 0) | n.s | .34 (lag 0) | .33 (lag 0) |
| 2. Number of news reports on COVID-19[a] | - | .55 (lag 0) | .91 (lag 0) | .56 (lag 0) | .36 (lag 2) |
| 3. Volume of mentions of COVID-19 on social media[a] | | - | .49 (lag 0) | .41 (lag 0) | .59 (lag 0) |
| 4. Volume of mentions of face masks on social media[a] | | | - | .35 (lag 0) | .41 (lag 0) |
| 5. Google search volume for COVID-19[a] | | | | | .49 (lag 4) |
| 6. Google search volume for face masks[a] | | | | | - |

[a] indicates the first series variable.

[b] indicates the second series variable

The value indicates the strongest significant cross-correlation coefficient at the .05 level between two variables at lag N. Positive/negative N indicates that the first series variable is a lagged indicator of the second series variable. If N is zero, it represents that the two variables are contemporaneously correlated.

of news reports, and the Google search volume from the three days before the official press release of the first confirmed case of COVID-19 in Taiwan (January 19) to the end of February. Since all six variables have autocorrelative relationships (S2 Appendix), the following outputs of CCF were partial correlation coefficients after controlling its own first-order autocorrelation.

The total number of confirmed cases of COVID-19 in Taiwan is correlated to the number of news reports on COVID-19 ($r$ = .34), volume of mentions of COVID-19 on social media ($r$ = .35), Google search volume for COVID-19 ($r$ = .34), and face masks ($r$ = .33) at lag 0, suggesting that the news media and the public immediately attached great importance to the imported cases of COVID-19 in Taiwan.

Moreover, the number of news reports on COVID-19 is correlated to volume of mentions of COVID-19 ($r$ = .55 at lag 0) and face masks ($r$ = .91 at lag 0) on social media and Google search volume for COVID-19 ($r$ = .56 at lag 0) and face masks ($r$ = .36 at lag 2). When the number of news reports on COVID-19 increases, the amount of discussion on social media and search behaviors on Google for COVID-19 and face masks increases, suggesting that nowadays, the information transmission from media to the public is rather quick, in fact almost instant. People who read the COVID-19-related information on the news could easily discuss it on the Internet via social media, which indirectly set the issue salience for the public. Moreover, with the increase in the number of people receiving the information of COVID-19, the Google search volume for COVID-19 on the same day and the Google search volume for face masks two days later on the Internet both increases. These results indicate that news media have a certain influence on the spread of the information of COVID-19 and indirectly impact what people think about and how they respond to COVID-19 during the epidemic period.

The trend chart Fig 1 provides another perspective on the dynamic movement of these time series variables. Fig 1 shows that when Taiwan's government announced its first confirmed case on January 21, it aroused the first wave of news media reports and attention to COVID-19 on social media, and prompted the public to search for COVID-19 and mask-related information on Google. After two more cases were added on January 24, the news media and the volume on social media climaxed again. The attention of the news media to COVID-19 continued to increase, as did the volume of discussion on COVID-19 on social media. Thus, social media rapidly promoted discussion of the topics in news reports, and there was a moderate cross correlation between the number of news reports and the volume of mentions of COVID-19 on social media ($r$ = .55 at lag 0). Additionally, since the spread of COVID-19 to Taiwan, its Google search volume peaked in less than five days (January 25; Fig 2), and the number of news reports and the volume of mentions of face masks on social media also peaked on January 28 when the first confirmed local case in Taiwan was identified as a household infection (Fig 1).

## A new form of public epidemic awareness

While the searching behaviors for COVID-19 and face masks are treated as an indicator of public epidemic awareness in the study, the meaning or the motive behind the two searching behaviors might be somewhat different. According to Fig 2, the trend chart showed that the rising period and time of reaching a peak were not the same (Fig 2). The statistical results also showed that the Google search volume for COVID-19 at time $t$ is correlated to the Google search volume for face masks at time t+2 ($r$ = .49), meaning that the increase in Google search volume for COVID-19 prompts the increase in the Google search volume for face masks two days later.

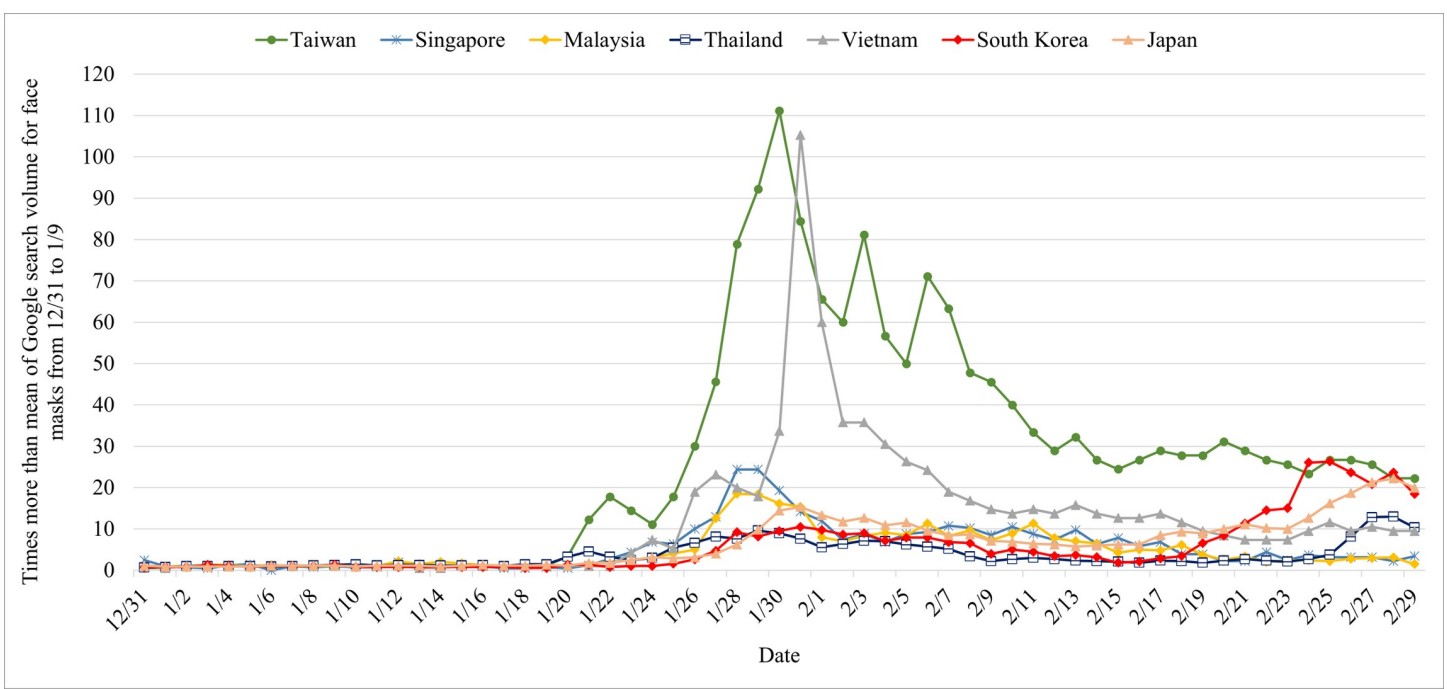

**Fig 1. The trend chart for the number of news reports on COVID-19, the volume of mentions of COVID-19 and face masks on social media, and total confirmed cases worldwide and in Taiwan between December 31, 2019 and February 29, 2020.**

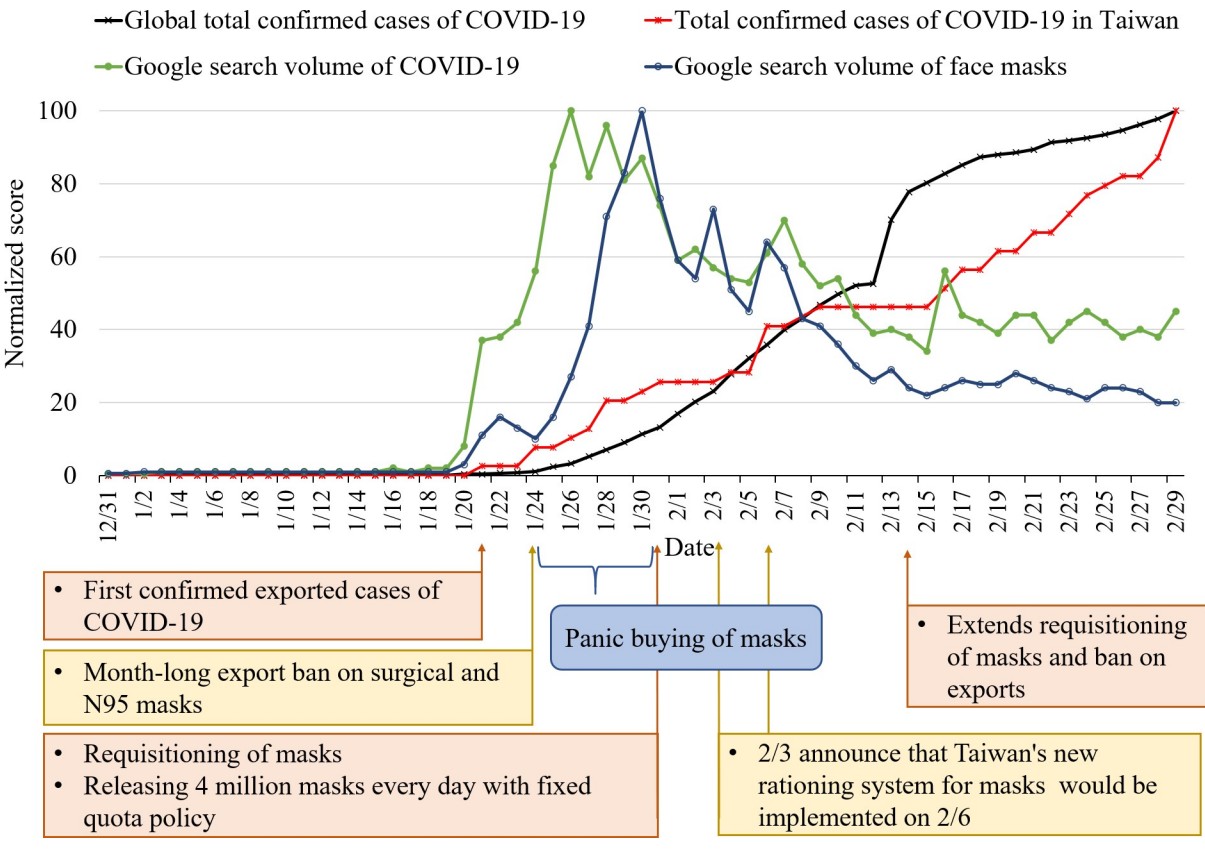

**Fig 2. The trend chart for Google search volume for COVID-19 and face masks, and the total confirmed cases worldwide and in Taiwan between December 31, 2019 and February 29, 2020.**

Figs 1 and 2 show that the volume of the mentions of face masks on social media and the Google search volume for face masks increased on January 24 after the government implemented an export ban on face masks, and reached peaks on January 28 and January 31, respectively. In this period, the government released 6 million epidemic prevention face masks for sale in convenience stores daily on January 28–30, with a purchase limit of three face masks per person at a price of NT$8 each. The second local case of household transmission occurred on January 31, and the WHO declared the outbreak a public health emergency of international concern on the same day. Furthermore, in the week of January 24–31, among the top 25 keywords of rising searches on Google Trends, nine were related to face masks, especially "medical face masks," "where to buy face masks," and "names of each face mask shop." This implies that public epidemic awareness began to increase in just a few days, from the perception of the importance of COVID-19 (Google search for COVID-19) to the perception of its threat and the adoption of protective measures (Google search for face masks).

It shows that new local cases, the government's emergency release of face masks, and lax purchase restrictions on face masks aroused a sense of insecurity and anxiety among the general public created by the increased chance of potential infection, and the public quickly began buying face masks. The phenomenon of panic buying of face masks was reported by news media during that period [43].

## Perspective of cross-national comparisons on the public epidemic awareness

Figs 3 and 4 depict the comparison of cross-national Google search volume for face masks and the number of total confirmed cases of COVID-19 in each country, respectively, among infected East and Southeast Asian countries, including Taiwan, Singapore, Malaysia, Thailand, Vietnam, South Korea, and Japan. Several findings can be drawn from the cross-national comparison. First, regardless of the countries, after the first confirmed case was announced, the Google search volume for face masks increased, and the search volume continued to grow along with the increased number of confirmed cases. This means that people concerned about the infectious disease and the public epidemic awareness was aroused in the earlier stage of the spread of COVID-19. Second, the extent of the increased search volume of face masks varies by country. It can be seen that the search volume of face mask in Taiwan during the spread of COVID-19 is dozens of times higher than the volume before the outbreak of COVID-19 (the base) and even more than a hundred times during the highest peak. The large difference in Google search volume before and after the outbreak of COVID-19 in Taiwan is obviously much higher than in the other East and Southeast Asian countries. The unique pattern of Google search volume for face masks provided further evidence that Taiwanese people regard face masks as important personal protective equipment for the prevention of COVID-19, and how Taiwanese people were worried about buying face masks in the initial phase of the spread of COVID-19.

However, it should be noted that when interpreting the Google search volume for face masks as public epidemic awareness in the context of cross-country comparison, some other factors varying by country affect people's face mask searching behaviors. For example, there is the cultural or national difference in the attitudes toward the use of face masks during the epidemic. Previous research has grouped countries such as Spain, Italy, the United Kingdom, Germany, and France into a no-mask wearing group, whereas countries such as Taiwan, Japan, and Thailand have been grouped into a mask-wearing group [44]. However, our study is not to advocate that the higher the Google search volume for face masks, the better the nation's epidemic control, but to provide another perspective on how Taiwanese perceive face mask as an important item of personal protective equipment for COVID-19.

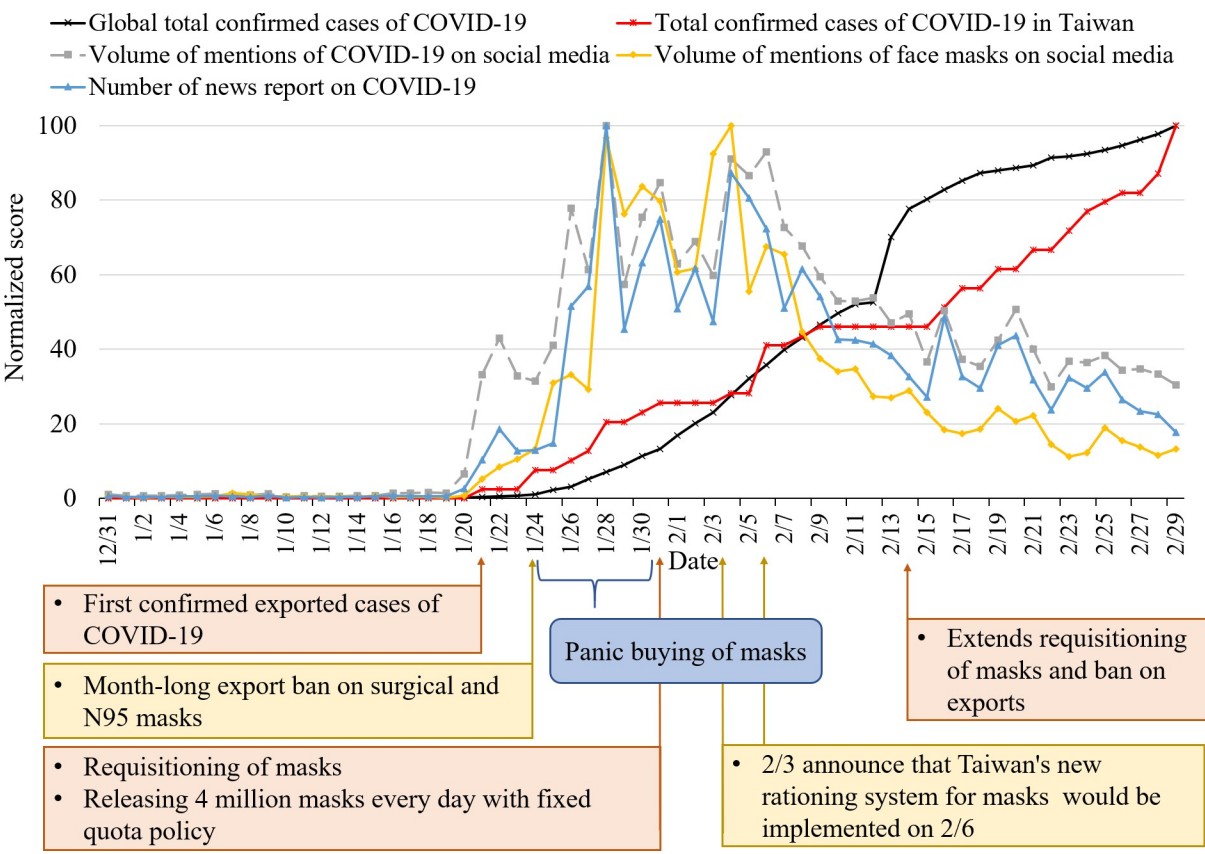

**Fig 3. Trends of Google search volume for face masks in Taiwan, Singapore, Malaysia, Thailand, Vietnam, South Korea, and Japan.**

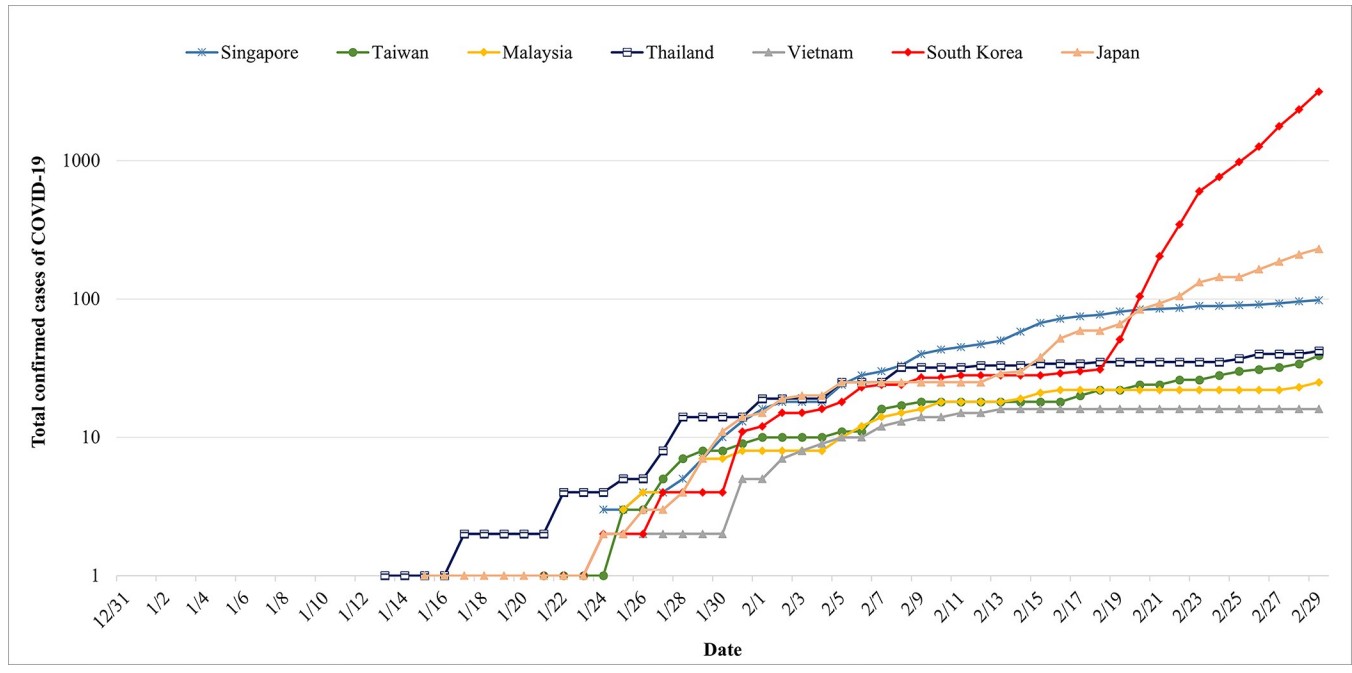

**Fig 4. Trends of total confirmed cases of COVID-19 in Taiwan, Singapore, Malaysia, Thailand, Vietnam, South Korea, and Japan.**

## Public opinion and a name-based rationing system for face masks

Public complaints also prompted the government to increase the allocation of resources for mask-related epidemic prevention materials. The government took emergency measures on January 31, including the daily collection of 4 million face masks from manufacturers, of which 2.6 million were for the public and were sold through convenience stores, drugstores, and related chain stores, with a purchase limit of three face masks per person per day at a unified price of NT$8 per face mask on February 1 [45]. However, online complaints about not being able to buy masks continued to appear, and people were rather dissatisfied with this government measure. Given that, the government considered the public opinion and subsequently launched a name-based rationing system for face masks on February 6. Based on the principles of prioritizing healthcare workers and ensuring equal purchase opportunities for all, the government adopted the method of unified collection, allocation, and price, and stipulated a purchase limit. Each person was allowed to buy two masks each week per National Health Insurance card (for identification) at a price of NT$5 per face mask [46]. This response measure was to ensure that the Taiwanese people would be protected from the hoarding of goods, and could avoid the more significant social burden caused by those who intended to drive up prices. After the policy was put in place, the government and the public worked together to create open data applications such as Face Mask Maps to enable the public to quickly find sales locations and inventory, thereby improving the efficiency of buying face masks, reducing the negative impact of long queues and the inability to buy face masks, and making proper use of information technology to achieve epidemic prevention [47].

After the implementation of the new mask rationing system, the government also successively put forward some face mask countermeasures such as increasing collection, extending the export ban, adding production lines, and increasing the purchasing limit of face masks to three per person per week [48–50]. This measure was expected to enable everyone to have face masks to protect themselves. Such positive actions reduced public anxiety about not being able to buy face masks; accordingly, the volume of the mentions and search query of face masks dropped significantly, which can be seen in Fig 1. A few weeks later, a "name-based rationing system for face masks new version 2.0" was implemented in which an online ordering mechanism was added on March 12 [51]. The purpose of this new mechanism was to ensure even better distribution and to make it more convenient to obtain face masks for people such as office workers and students who lack time to go to pharmacies and public health centers. From the perspective of crisis management, these face mask countermeasures not only achieve a cogent allocation of medical resources and reduce the possibility of infection, but also take into account the panic of the public so that people have sufficient control over the purchase of face masks.

Additionally, it is worth mentioning that while the Google search volume for face masks in Taiwan declined after the announcement and implementation of face mask related countermeasures, the Google search volume for face masks is still 20 to 30 times higher than the search volume before the epidemic spread, and has been maintained for a period of time (Fig 3). These values are mostly more elevated than the values of other countries (Fig 3). It shows that the government's face mask countermeasures have not only reduced public anxiety, which caused a tremendous search volume for face masks, but also played a role in maintaining the public epidemic awareness. It should be noted that keeping the public epidemic awareness at a high level might be crucial for combating COVID-19. Our study found that when the Google search volume for face masks gradually reduced to a relatively low point in Korea (February 17) and Japan (February 15), the "re-outbreak" of the COVID-19 epidemic appeared with large-scale confirmed cases.

**Table 2. The result of one-way analysis of variance.**

|  | Sum of squares | df | Mean square | F | p |
|---|---|---|---|---|---|
| Between groups | 6163.83 | 5 | 1232.77 | 5.67 | < .05 |
| Within groups | 76726.75 | 353 | 217.36 |  |  |
| Total | 82890.58 | 358 |  |  |  |

### The period of highest Google search volume for face masks and the loss of control

According to the ANOVA results (Table 2), the types of fear-embedded message, had a significant effect on the Google search volume for face masks [$F(3,358) = 5.67$, $p < .05$]. It means that the Google search volume for face masks varied by the dissemination of various types of fear-embedded message. Specifically, the results from post hoc comparisons using the Scheffé test with the bootstrap method (Table 3) indicated that only the mean score for the *loss of control* group ($M = 42.65$, $SD = 26.12$) was significantly different from the other groups ($M = 26.73$, $SD = 14.61$, for *mistrust*; $M = 24.65$, $SD = 18.38$, for *severity*; $M = 22.08$, $SD = 21.20$, for *uncertainty*; $M = 26.12$, SD = 12.10, for *susceptibility*; $M = 25.77$, $SD = 8.67$, for *without fear*).

### Changes in the number of different fear events at different periods

A Pearson chi-square test was performed to examine the change in the proportion of events involving different types of fear across various time periods. The proportion differs by time period [$\chi2 (10) = 41.16$, p < .05]. A cross-table (Table 4) shows the counts and expected counts of events in different categories.

To further ascertain which types of events in terms of fear have relatively large ratios compared to different periods, multinomial logistical regression was used (Table 5). We used the second stage as a reference group since it had the highest Google search volume for face masks. The odds ratio for *uncertainty* (6.90) in the First stage is significant, indicating that the probability of an *uncertainty* event happening in the First stage is 6.90 times higher than that in the Second stage, relative to the *without fear* group. On the other hand, the probability of the *loss of control* event happening in the Second stage is 3.23 (1/0.31) times higher than that in the Third stage, relative to the *without fear* group.

**Table 3. Post hoc Scheffé test with the bootstrap method.**

|  | Descriptive statistics | | | Multiple comparison [a] | | | | |
|---|---|---|---|---|---|---|---|---|
|  | N | Mean | SD | 2[b] | 3[b] | 4[b] | 5[b] | 6[b] |
| 1. Mistrust | 48 | 26.73 | 14.61 | 2.08 (3.31) | **-15.92* (3.92)** | 4.65 (3.18) | 0.62 (2.49) | 0.96 (2.65) |
| 2. Severity | 34 | 24.65 | 18.38 | - | **-18.00* (4.16)** | 2.57 (3.46) | -1.47 (2.84) | -1.12 (2.98) |
| 3. Loss of control | 20 | 42.65 | 26.12 |  | - | **20.57* (4.06)** | **16.54* (3.54)** | **16.88* (3.66)** |
| 4. Uncertainty | 39 | 22.08 | 21.20 |  |  | - | -4.04 (2.69) | -3.69 (2.84) |
| 5. Susceptibility | 131 | 26.12 | 12.10 |  |  |  | - | 0.34 (2.04) |
| 6. Without fear | 87 | 25.77 | 8.67 |  |  |  |  | - |

Values in parentheses are bootstrap standard error.

[a]Post hoc Scheffé test with the bootstrap method was used.

[b]The reference group for the multiple comparisons of mean.

*The mean difference in bold is significant at the .05 level.

**Table 4. A cross-table for sample characteristics with Chi-square analysis.**

| | | | Time periods | | | $\chi^2$ (df) |
|---|---|---|---|---|---|---|
| | | | First stage (N = 47) | Second stage (N = 105) | Third stage (N = 207) | |
| Types of fear | Untrustworthy (N = 48) | n | 6 | 17 | 25 | 41.16* (10) |
| | | Expected n | 6.3 | 14.0 | 27.7 | |
| | Dread (N = 34) | n | **8**[a] | 11 | 15 | |
| | | Expected n | 4.5 | 9.9 | 19.6 | |
| | Loss of control (N = 20) | n | 2 | **10**[a] | 8 | |
| | | Expected n | 2.6 | 5.8 | 11.5 | |
| | Uncertainty (N = 39) | n | **15**[a] | 10 | *14*[b] | |
| | | Expected n | 5.1 | 11.4 | 22.5 | |
| | Vulnerability (N = 131) | n | *11*[b] | 34 | 86 | |
| | | Expected n | 17.2 | 38.3 | 75.5 | |
| | Without fear (N = 87) | n | *5*[b] | 23 | 59 | |
| | | Expected n | 11.4 | 25.4 | 50.2 | |

[a]Counts at least 1.5 times greater than expected counts are in **bold**.

[b]The counts that are 1.5 times less than the expected counts are in *Italics*.

*The Pearson chi-square value is significant at the .05 level.

## Discussion

The current situation of the COVID-19 pandemic is serious. This is having a dramatic impact on the health of citizens around the world, and is making it more difficult for governments to prevent the spread of the disease. When epidemic prevention and the economy stand at opposite ends of the scale, finding a balance is the biggest challenge for governments. This study provides an insight into how to measure national public epidemic awareness through social media- and Internet-based data, and the role the public plays in epidemic prevention and control.

**Table 5. Results of the multinomial logistic model.**

| Time periods [a] | | B | SD | Wald | df | Odds ratio | 95% Confidence Interval for odds ration | |
|---|---|---|---|---|---|---|---|---|
| | | | | | | | Lower Bound | Upper Bound |
| First stage | Intercept | -1.53 | 0.49 | 9.56 | 1 | .00 | | |
| | Mistrust | 0.48 | 0.68 | 0.50 | 1 | .48 | 1.62 | .42 |
| | Severity | 1.21 | 0.68 | 3.17 | 1 | .07 | 3.35 | .89 |
| | Loss of control | -0.08 | 0.92 | 0.01 | 1 | .93 | .92 | .15 |
| | Uncertainty | 1.93* | 0.64 | 9.10 | 1 | .00 | 6.90 | 1.97 |
| | Vulnerability | 0.40 | 0.60 | 0.43 | 1 | .51 | 1.49 | .46 |
| | Without fear [b] | - | - | - | - | - | - | - |
| Third stage | Intercept | 0.94 | 0.25 | 14.69 | 1 | .00 | | |
| | Mistrust | -0.56 | 0.40 | 1.94 | 1 | .16 | .57 | .26 |
| | Severity | -0.63 | 0.47 | 1.83 | 1 | .18 | .53 | .21 |
| | Loss of control | -1.17* | 0.53 | 4.76 | 1 | .03 | .31 | .11 |
| | Uncertainty | -0.61 | 0.48 | 1.58 | 1 | .21 | .55 | .21 |
| | Vulnerability | -0.01 | 0.32 | 0.00 | 1 | .96 | .99 | .53 |
| | Without fear [b] | - | - | - | - | - | - | - |

[a,b]The reference categories are *Second stage* for time periods and *Without fear* for types of risk characteristic.

* The parameter estimate is significant at the .05 level.

## The role of news media and government in rapidly increasing public epidemic awareness in the early stage of the spread of COVID-19

When it comes to epidemic spread, the basic reproduction number ($R_0$) is a marker of epidemic transmission intensity under a specific spatiotemporal background. This number represents the average number of secondary infections produced by a typical case of an infection. During the early stages of an epidemic, scientists estimate that if $R_0$ is less than 1.5, in theory, contact tracing of patients can be used to identify potential targets for control. However, during the actual outbreak of the COVID-19 epidemic, controlling it was difficult because of the substantial increase in the number of patients and asymptomatic transmission [52]. $R_0$ was found to be more than 2 in Wuhan during the early stages of the epidemic, and in Europe and the Americas from March onwards [53, 54]. This shows that it is essential to lower $R_0$ in the early stage of the epidemic spread. As of March 31, the data showed that the total number of confirmed cases of COVID-19 per million in Taiwan is considerably lower than in most other countries [6] even though Taiwan resisted the COVID-19 epidemic for more than 70 days and has high population density (673 people/$km^2$). The $R_0$ of Taiwan was still lower than 1, and the epidemic was within the controllable range at the end of March [53]. This might be attributed to the government's super-early deployment [3] as well as the early rise of public epidemic awareness.

Previous studies suggested the earlier and the louder the news media reports disease-related messages, the better it could help the public to prevent infection [13, 14]. Our study shows that the media in Taiwan did these two things in the early stage of COVID-19 and successfully increase public epidemic awareness. Moreover, the role of government communication cannot be neglected when it comes to the rise of public epidemic awareness. In Taiwan, the government put much effort into public communication. For example, the Central Epidemic Command Center (CECC) has been holding a daily press briefing since the first case of COVID-19 was confirmed. The CECC releases the latest results of the epidemic, announces the countermeasures against the COVID-19 epidemic, advocates good hygiene, fights against the infodemic, and allows journalists to ask questions about COVID-19. For governments, it seems that communicating COVID-19 information in a transparent, accurate, and timely manner might be helpful for epidemic control. It is because it gives the media and the public opportunities to get more and correct information about how to combat COVID-19. Meanwhile, it also provides a space for the public to discuss the appropriateness of the government's countermeasures such as the policy related to face masks as discussed in this study. Perhaps, such a serious and prompt way that the government communicated the COVID-19 information and mitigation strategies to the public via media is the reason why the large differences in Google search volume for face masks before and after the outbreak of COVID-19 appears in Taiwan but not the same level in other Asian counties.

In terms of disease information-seeking behavior, previous findings suggested that seeking behaviors for epidemic diseases could be an indicator of influenza outbreaks [27]. However, the relationship varied by country [28]. We argue that this variation might be explained by differences in public epidemic awareness. In Taiwan, the public's COVID-19 online information-seeking behaviors appeared quite early, with high query volumes. When the media began to report on the epidemic on a large scale, the discussion of COVID-19 continued to increase on social media, and the public started seeking information about COVID-19 through Google to reduce their sense of insecurity caused by uncertainty about the unknown. Such rapid growth of public epidemic awareness is immensely significant for public health. Realizing the importance of COVID-19 and understanding COVID-19 is the crucial first step towards combating it.

## The story behind the face mask information-seeking behavior during the epidemic

While the search for diseases can be viewed as a form of public epidemic awareness [29], we argue that the behavior motivations of information-seeking for COVID-19 and face masks differ. Driving the search for information about the disease was the public's feeling of ignorance in the initial course of the spread of the emerging infectious disease. They sought knowledge to reduce their uncertainty and to understand its potential threat. By contrast, face masks are crucial personal protection equipment [35], and driving their search was the public's perception of the threat of COVID-19 as well as a desire to understand how to effectively protect themselves. In Taiwan, the rapid growth in the Google search volume for face masks occurred when the Google search volume for COVID-19 reached its highest peak. It shows that after understanding the threat of COVID-19 via searching online, people started to find face masks to protect themselves from being infected. Our finding indicates that the difference in the number of Google search volume for face masks before and after the outbreak of COVID-19 is extremely large, and the value in Taiwan is higher than that in the other infected East and Southeast Asian countries. It seems that Taiwanese people became more active than usual in seeking face masks online after the spread of COVID-19. The initial massive searches for face masks might be explained by the miserable memories of SARS in Taiwan [55]. At that time, there were 346 confirmed SARS cases, and 73 SARS patients died [56].

Despite the ongoing debates on the effectiveness of face masks in epidemic transmission [35, 57, 58], several studies have shown that face masks could prevent transmission of human coronavirus [59, 60]. Previous research found that people wearing a face mask tended to have positive hygiene practices such as maintaining social distancing, washing hands as well as avoiding crowds, and regularly avoiding close contact with an infected person [36]. Thus, the face mask information-seeking behavior might be viewed as an indicator of public epidemic awareness.

While our data could not directly reflect one's actual preventive behavior of wearing a face mask and prove its effectiveness of epidemic prevention, previous literature has documented that Taiwan adopted widespread measures for the public to wear masks would be the reason for the reduction of COVID-19 transmission and even having good control of the COVID-19 without a mandatory suspension of work and school [61]. For example, TCDC requires people to wear masks in public places, including public transport and hospital, and that will be fined if they fail to comply. The other research also showed that Taiwan, a mask-wearing country, has lower growth rates of COVID-19 cases compared to the non-mask-waring counties in the early stage of the spread of COVID-19 via combining mathematical modeling and existing scientific evidence [44].

## The importance of the maintenance of public epidemic awareness from the cross-national perspective

Regarding the face mask information-seeking behaviors, our findings suggested that Google searches for face masks rapidly increased in the outbreak of COVID-19 in East and Southeast Asian countries, including Taiwan, Singapore, Malaysia, Thailand, Vietnam, South Korea, and Japan, which is similar to previous studies [57, 58]. A prior study [49] suggested that while the early rise of public epidemic awareness might play a role in preventing the spread of epidemics, how to maintain public epidemic awareness is the key to defeating COVID-19 [55]. This study provides examples of Italy, Spain, and South Korea as evidence to describe how a decline in Google search volume for face masks could lead to the "re-outbreak" of COVID-19. Our study also found a similar pattern in South Korea and Japan. It implies that if people start to ignore the potential threat of COVID-19, it would reappear.

### Types of fear-embedded popular events and public epidemic awareness in the course of COVID-19 spread in Taiwan

In the history of infectious diseases, fear is almost always humans' first intuitive emotion and response [23]. Among the five types of fear in this study, the most common fear factor embedded in popular events across the whole period was found to be *susceptibility*. The content of these events includes new cases throughout the world and the events in which people who did not obey the rules of home quarantine. From changes in the proportion of various types of fear in popular COVID-19 topics over time, different fear elements play different roles in different time periods. This also properly shows the public response to COVID-19.

Before COVID-19 entered Taiwan, and up until the first case occurred in Taiwan (First stage: 31 December 2019 to 25 January 2020), the types of fear embedded in popular events that were discussed by the public were mainly the *uncertainty* and *severity* of COVID-19. The number of these types of events was 1.5 times higher than the expected value. The event content included discussion on the possibility of human-to-human transmission of COVID-19 and where suspected patients were located. Additionally, as studies during this period provided that COVID-19 is caused by the severe acute respiratory syndrome coronavirus two, the public in Taiwan was concerned about the possibility of severe harm or even death, similar to what was caused by SARS in the past. These uncertain events and consideration of the immense threat of COVID-19 to individual health sowed the seeds of fear in the population for days to come.

As the epidemic gradually spread, the number of popular events related to *loss of control* started to increase in the second stage (26 January to 21 February). These types of events accounted for a higher proportion compared to other stages. Since the second stage features high Google search volume for face masks, we argue that the *loss of control* might explain why panic behavior and the huge amount of searching behaviors for face masks occur in Taiwan, as it involves social and livelihood issues. This causes individuals to lose control over self-protection and to feel uneasy when they are unable to buy masks and other essential supplies for epidemic protection and living. Thus, it gives us an insight that when facing infectious disease, the issue of achieving a cogent allocation of medical resources is important for the government to avoid causing excessive public panic.

Several limitations of the study ought to be mentioned here. First, the Google search volume for "face masks" did not directly reflect the actual behavior of wearing face masks. To some extent, the Google search volume represents the public perception of the COVID-19 threat and an attempt to take self-protection measures. Second, while our results suggest that the Google search volume for face masks could be an indicator of public epidemic awareness, the indicator might not be applicable to every country due to cultural differences in attitudes toward using face masks during the epidemic [11, 44]. Finally, our research used the web methodology to collect timely information about public epidemic awareness and the quantity of disease-information spread on the Internet and examine the dynamic relationships. However, we did not consider the content of information transmitted on the Internet. Future studies might adopt various analytics such as text mining and sentiment analysis to take what the news spread into account.

## Conclusions

Facing the threat of COVID-19, the government's crisis risk management and public epidemic awareness are at the core of this campaign. In addition to the government's early maneuvers and precautions, the study highlights that the importance of early public vigilance against COVID-19 and the timely detection of public epidemic awareness are needed for effective

epidemic prevention. The news media also play a role in disseminating information between the government and the public. The volume of mentions and search query behaviors on social media has many implications in the anti-epidemic stage. These online behaviors can be regarded as indicators for the government to observe public epidemic awareness and for the public to monitor the government's response to crisis risk management. However, the study recognized that using the search behavior of face masks as an indicator of public epidemic awareness cannot be used in all countries. Each country or culture has its own attitudes toward the use of face masks or its face masks-related policy. However, we still believe that finding out an appropriate digital footprint of public epidemic awareness in a given scenario for a region or a country is necessary because it provides real-time information on public awareness of the epidemic. Finally, while fear is an inevitable product when an emerging infectious disease occurs, by understanding why fear arises among the public and which fear factors cause excessive public panic, the government and society can convert resistance into assistance. In a society where the Internet and science and technology are booming, the application of big data in the fight against epidemics will be more diverse in the future.

## Supporting information

**S1 Appendix. Information about the scope of the data and KEYPO big data analytics engine.**
(DOCX)

**S2 Appendix. The output of the autocorrelation function (ACF) and partial autocorrelation function (PACF).**
(DOCX)

**S3 Appendix. The output of the Cross-Correlation Function (CCF) after whitening the data by removing the first-order autocorrelation.**
(DOCX)

**S1 File.**
(XLSX)

**S2 File.**
(XLSX)

**S3 File.**
(XLSX)

**S1 Table. The categorization of the types of fear.**
(DOCX)

**S2 Table. The results of one-way analysis of variance for Google search volume for face masks in different periods.**
(DOCX)

**S3 Table. The results of post hoc multiple comparisons for Google search volume for face masks in different periods.**
(DOCX)

## Author Contributions

**Conceptualization:** Chih-Yu Chin.

**Data curation:** Chih-Yu Chin, Cheng-Lung Wang.

**Formal analysis:** Chih-Yu Chin, Cheng-Lung Wang.

**Funding acquisition:** Chih-Yu Chin.

**Methodology:** Chih-Yu Chin, Cheng-Lung Wang.

**Supervision:** Chih-Yu Chin, Chang-Pan Liu.

**Visualization:** Cheng-Lung Wang.

**Writing – original draft:** Chih-Yu Chin.

**Writing – review & editing:** Chih-Yu Chin, Chang-Pan Liu, Cheng-Lung Wang.

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
