## [Decision Letter · Decision Letter 0]

7 Oct 2020

PONE-D-20-17953

Evolving public behavior and attitudes towards COVID-19 and face masks in Taiwan: A social media study

PLOS ONE

Dear Dr. Chin,

Thank you for submitting your manuscript to PLOS ONE. After careful consideration, we feel that it has merit but does not fully meet PLOS ONE’s publication criteria as it currently stands. Therefore, we invite you to submit a revised version of the manuscript that addresses the points raised during the review process.

The reviewers suggest: (1) to make sure the number of searches does not necessarily reflect the number of individuals searching for the item; (2) to discuss about the search trends in other South East Asian countries.

In light of the reviewers' suggestions, I invite you to make a revision. 

We look forward to receiving your revised manuscript.

Kind regards,

Chang Sup Park, Ph.D.

Academic Editor

PLOS ONE

Journal Requirements:

2.Thank you for stating the following in the Financial Disclosure section:

[This manuscript was supported by the Ministry of Science and Technology, Taiwan, under grant number MOST 108-2410-H-033-043. The corresponding author, CY, is the principal investigator of the research project.

The URL of funder website is https://www.most.gov.tw/?l=en.

Finally, the funder had no role in study design, data collection and analysis, decision to publish, or preparation of the manuscript.].   

We note that one or more of the authors are employed by a commercial company: Big Data Co., Ltd.,

Additional Editor Comments (if provided):

The reviewers suggest: (1) to make sure the number of searches does not necessarily reflect the number of individuals searching for the item; (2) to discuss about the search trends in other South East Asian countries.

In light of the reviewers' suggestions, I invite you to make a revision.

Reviewers' comments:

Reviewer's Responses to Questions

**Comments to the Author**

1. Is the manuscript technically sound, and do the data support the conclusions?

Reviewer #1: Yes

2. Has the statistical analysis been performed appropriately and rigorously? 

Reviewer #1: Yes

3. Have the authors made all data underlying the findings in their manuscript fully available?

Reviewer #1: Yes

4. Is the manuscript presented in an intelligible fashion and written in standard English?

Reviewer #1: Yes

5. Review Comments to the Author

Reviewer #1: a. This is an interesting study, that used search trends using terms related to face mask in google data base. The number of searches in the google database, not necessary a reflection of number of individuals searching for that item. A particular person, who needs an item, may search it several times on the google database and it adds to the number of searches on that particular time period. This way the google trends may not provide accurate information.

b. The authors may discuss about the search trends related to face mask during the defined period in other countries of the SE Asian region. This will give an idea that Taiwan is successful in containing the panic buying behavior through its policy, whereas other countries, were unable to do so.

6. PLOS authors have the option to publish the peer review history of their article (what does this mean?). If published, this will include your full peer review and any attached files.

Reviewer #1: **Yes: **Sujit Kumar Kar

---

## [Author Response · Author response to Decision Letter 0]

1 Nov 2020

Response letter for revised Manuscript PONE-D-20-17953R1 

“Evolving public behavior and attitudes towards COVID-19 and face masks in Taiwan: A social media study”

Authors: Chih-Yu Chin, Chang-Pan Liu, Cheng-Lung Wang

We are sincerely grateful to the reviewer for constructive comments. We believe that these comments have helped us enhance the quality of the manuscript. We have done our best to revise as well as improve the paper according to the comments. Please see our responses to each comment as follows.

Comment #1: "This is an interesting study, that used search trends using terms related to face mask in google data base. The number of searches in the google database, not necessary a reflection of number of individuals searching for that item. A particular person, who needs an item, may search it several times on the google database and it adds to the number of searches on that particular time period. This way the google trends may not provide accurate information."

Response #1:

Thank you for appreciating our effort in measuring the public epidemic awareness by the behaviors of online searches for face masks. We are also grateful to the reviews for this helpful comment. 

We have checked that Google Trend would eliminate repeated inquiries from the same person over a short period regarding duplicate searches. We have added this description to the revised version of the paper. Please refer to line 188-190 of page 11. 

Comment #2: "The authors may discuss about the search trends related to face mask during the defined period in other countries of the SE Asian region. This will give an idea that Taiwan is successful in containing the panic buying behavior through its policy, whereas other countries, were unable to do so."

Response #2:

We are very thankful to the reviewer for this helpful and constructive suggestion. In this revised version of the paper, we have discussed the cross-national comparison of Google search volume of face masks, and the total confirmed cases of COVID-19.

As for country selection, we not only discuss the infected Southeast Asian countries as suggested by the reviewer but also discussed the infected countries, which are the same as Taiwan in East Asia. Therefore, data from seven countries, including Taiwan, Singapore, Malaysia, Thailand, Vietnam, South Korea, and Japan, were collected to make a cross-national comparison in the study. These countries shared common conditions that had resisted the COVID-19 epidemic for more than 30 days from the date that the countries identified the first confirmed case of COVID-19 to February 29, 2020. Moreover, the number of total confirmed cases of COVID-19 is greater than 15 on February 29, 2020. please refer to lines 193-199 of page 11.

Moreover, the cross-national comparison findings did provide another perspective on how Taiwan might be different from other countries. Moreover, when comparing various countries simultaneously, we can better understand the universal and unique patterns of online searches for face masks across nations. We have added several paragraphs in a proper section. Please refer to line 316-336 of pages 18 and 19, line 383-395 of pages 22 and 23, line 477-489 of pages 31 and 32, and 497-510 of pages 32 and 33.

---

## [Decision Letter · Decision Letter 1]

6 Jan 2021

PONE-D-20-17953R1

Evolving public behavior and attitudes towards COVID-19 and face masks in Taiwan: A social media study

PLOS ONE

Dear Dr. Chin,

Thank you for submitting your manuscript to PLOS ONE. After careful consideration, we feel that it has merit but does not fully meet PLOS ONE’s publication criteria as it currently stands. Therefore, we invite you to submit a revised version of the manuscript that addresses the points raised during the review process.

Reviewers acknowledge that the revised manuscript improved somewhat compared with the original submission. But they still have lingering concerns.

They point out that you need to create theoretically more justifiable hypotheses and take the time series nature of the data more seriously. They also suggest you mention all the methods you used in the analysis. I concur.

For your information I attach the reviewer comments at the bottom of this email.  I hope you will find them to be constructive and helpful, in revising the manuscript.

We look forward to receiving your revised manuscript.

Kind regards,

Chang Sup Park, Ph.D.

Academic Editor

PLOS ONE

Additional Editor Comments (if provided):

Reviewers acknowledge that the revised manuscript improved somewhat compared with the original submission. But they still have lingering concerns.

They point out that you need to create theoretically more justifiable hypotheses and take the time series nature of the data more seriously. They also suggest you mention all the methods you used in the analysis. I concur.

For your information I attach the reviewer comments at the bottom of this email. I hope you will find them to be constructive and helpful, in revising the manuscript.

Reviewers' comments:

Reviewer's Responses to Questions

**Comments to the Author**

1. If the authors have adequately addressed your comments raised in a previous round of review and you feel that this manuscript is now acceptable for publication, you may indicate that here to bypass the “Comments to the Author” section, enter your conflict of interest statement in the “Confidential to Editor” section, and submit your "Accept" recommendation.

Reviewer #1: All comments have been addressed

Reviewer #2: (No Response)

2. Is the manuscript technically sound, and do the data support the conclusions?

Reviewer #1: Yes

Reviewer #2: No

3. Has the statistical analysis been performed appropriately and rigorously? 

Reviewer #1: Yes

Reviewer #2: No

4. Have the authors made all data underlying the findings in their manuscript fully available?

Reviewer #1: Yes

Reviewer #2: (No Response)

5. Is the manuscript presented in an intelligible fashion and written in standard English?

Reviewer #1: Yes

Reviewer #2: Yes

6. Review Comments to the Author

Reviewer #1: This is an interesting article. The authors had responded to all the queries raised. Revision made is satisfactory

Reviewer #2: The manuscript “Evolving public behavior and attitudes towards COVID-19 and face masks in Taiwan: A social media study” presents a correlational analysis of COVID-19 and mask related search behaviour in Taiwan and South East Asian countries as well as a content analysis. I was given this manuscript to review after a round of revision with a different reviewer. Normally at this stage I simply address whether or not the authors have addressed the reviewers' concerns in a satisfactory manner. But it seems very little substantive ground was covered in the previous round. I have some serious misgivings about the manuscript as currently constituted.

Theory

I don’t have a clear sense of the argument the authors are advancing or the hypotheses they are testing. At various points the authors make claims about the effects of awareness on pandemic mitigation, but of course their design is not set up to test that possibility. Is it that COVID-19 cases caused awareness, and then mask usage for prevention?

Relatedly, what are the expectations related to fear and the outcomes? I get that fear may be associated with awareness and mask usage, but the authors construct some typology of fear without any clear purpose in mind. All of this needs to be much more strongly motivated by theory. I had no clear sense of what expectations the authors wanted to test, so the rest of the manuscript felt like a fishing expedition.

The authors make an assumption that mask usage (or searching) can act as an indicator that the public is taking steps to protect themselves. I understand that is a reasonable assumption in Taiwan. It might be worth discussing the variation that exists on this dimension cross-nationally. We wouldn’t be able to use mask searching or usage as such an indicator when public health authorities actively dismissed their usage in many other countries until several months into the pandemic (see Merkley & Loewen, 2020).

Which brings me to a much more substantive point: government communications are notably absent from this analysis. As noted, I’m not entirely clear on the argument advanced here. But it seems like there is an assumption that cases generated media and public awareness, and subsequently searching related to masks. Of course, cases increased in other Asian countries, but not the same level of searching for masks, which could be in large part due to the seriousness and promptness with which their governments acted on the pandemic and the mitigation strategies they communicated to their mass publics. I think there would be a lot of value added to the analysis if there was some effort to identify a link between government communication and media/the public, along the lines of what we see in agenda setting literature (see Soroka 2002 and many others along those lines).

Data and Method

My biggest concerns relate to the data and methods. First, much more information needs to be given about the precise nature of the data being used in this manuscript, especially related to social media. We need more information on what data KEYPO collects and from what platforms. “Social media” is a very broad term. What exactly is the structure of the data that is provided? How is it accessed? What keywords are used? What are its limitations? None of this is transparently discussed in the manuscript.

- A minor, but related point. Wikipedia is not an acceptable source for students, and is not an acceptable source for published research. These data need to be verified from the original source

Second, I do not think the authors are taking the time series nature of their data seriously enough. There is temporal dependence between these data that do not lend themselves well to simple correlations. For example, confirmed cases will be mechanically related to prior values (in fact, these variables are unit root by definition), while others, like social media and news reports may be (trend) stationary. Autocorrelation likely also needs to be accounted for. All of their series display evidence of trending with a clear structural break when COVID-19 arrived on the scene. I see no reason to see these correlations as evidence of anything other than co-trending. I would expect a linear trend would also be strongly correlated with cases and the other indicators used here. The authors need to estimate models that meet the assumptions of time series analysis.

Third, the authors need to make a stronger case for using social media data to infer behaviour. We know that online behaviour and social media usage is occurring among non-representative segments of the population (i.e. younger, more educated). An alternative strategy would be to survey individuals over the course of the pandemic. Other scholars have done this, so it cannot be dismissed out of hand because it is “hard” (see Merkley & Loewen, 2020; Sides, Tausanovitch and Vavreck 2020; Clinton et al. 2020 Science Advances, etc.). More validation is needed to justify using social media data as a proxy for public opinion or at least a more thorough discussion of the limitations.

Why is the time frame for stage 1 different from stage 2? The content analysis stretches an entire month longer. Why not collect more data over a longer time frame? Why does N=38 in Table 1 if data is collected over two months

There is not nearly enough detail on the content analysis. What does it mean that KEYPO provides samples of “popular events”? Are these article headlines? Who coded each event as to whether they met criteria for inclusion in the fear based categories? What were these fear dimensions chosen, and what does each represent? What is the coding scheme? How have the authors validated the reliability of the coding?

Structure and Contribution

The authors need to be much more attentive to limiting the claims they make to what the simple correlational data they present can sustain. Their research design lacks causal identification or even any strategy to make such claims plausible. These are simple bivariate correlations with small N. They cannot use this data to sustain an argument that public awareness in Taiwan mitigated the pandemic.

The paper was hard to follow. The authors cycle back and forth between discussion of results and commentary on Taiwan’s governmental response to COVID-19. Certain methods and analysis were also not introduced in the relevant section, so they appeared as a surprise when describing the results. This should be revised and a larger effort at signposting is needed throughout to guide readers step by step through the manuscript.

7. PLOS authors have the option to publish the peer review history of their article (what does this mean?). If published, this will include your full peer review and any attached files.

Reviewer #1: **Yes: **Sujit Kumar Kar

Reviewer #2: No

---

## [Author Response · Author response to Decision Letter 1]

3 Feb 2021

Response letter for revised Manuscript PONE-D-20-17953R1 

“Evolving public behavior and attitudes towards COVID-19 and face masks in Taiwan: A social media study”

Authors: Chih-Yu Chin, Chang-Pan Liu, Cheng-Lung Wang

We are sincerely grateful to the reviewers for showing the approval of our manuscript and providing constructive comments. We believe that these comments have helped us enhance the quality of the manuscript. We also have done our best to revise as well as improve the paper according to the comments. Please see our responses to each comment as follows.

Reviewer 1’s comments

Comment #1 (Reviewer 1): This is an interesting article. The authors had responded to all the queries raised. Revision made is satisfactory. 

Response #1:

Thank you for appreciating our effort in revising our previous work. We are also grateful to those constructive comments which definitely improve the quality of our manuscript. 

Reviewer 2’s comments

Comment #1: 

I don’t have a clear sense of the argument the authors are advancing or the hypotheses they are testing. At various points the authors make claims about the effects of awareness on pandemic mitigation, but of course their design is not set up to test that possibility. Is it that COVID-19 cases caused awareness, and then mask usage for prevention?

Relatedly, what are the expectations related to fear and the outcomes? I get that fear may be associated with awareness and mask usage, but the authors construct some typology of fear without any clear purpose in mind. All of this needs to be much more strongly motivated by theory. I had no clear sense of what expectations the authors wanted to test, so the rest of the manuscript felt like a fishing expedition.

Response #1:

We are very thankful to the reviewer for this helpful and constructive suggestion. In this revised version of the paper, we have re-construct the section of Introduction to make our contentions more understandable for readers. At the initial stage of the COVID-19 epidemic, Taiwan has performed a good epidemic control. However, previous research on Taiwan’s epidemic prevents seemed to discuss what measures the government has done. The study, building on previous research emphasizing the effectiveness of earlier public epidemic awareness on epidemic prevention, provides another story about how public epidemic awareness rose and how it was associated with epidemic development, media, and government in modern society during the development of the COVID-19 epidemic. Therefore, in this reversion, we have put more emphasis on discussing the role of media and government in the rise of public epidemic awareness. Please refer to line 73-161 from page 4 to page 9.

Moreover, regarding the reviewer’s question of what are the expectations related to fear and the outcomes, we have revised the relevant paragraphs in the section of Introduction and Methods to make our expectations related to fear and outcomes more clear. 

In the section of Introduction, first of all, we emphasized that not only the quantity of COVID-19 related message but also the contents of the message would have an impact on the public opinion formation of COVID-19. Then, we provided theoretical and empirical evidence to emphasize the role of fear in rising public epidemic awareness and health behavior change. Finally, we acknowledged that, in this case, using fear appeals might be “The same knife cuts bread and fingers.” Thus, we then argued that understanding messages associating with various characteristics of fear spread to people and its effect on public epidemic awareness during the epidemic would be helpful for the government to enact an appropriate strategy to arouse and citizen’s fear or mitigate the panic. Thus, the study aims to explore the dynamic relationships between epidemic development and the disease-related information spread on the Internet and to explain how it contributes to the epidemic control during the early period of the COVID-19 epidemic in Taiwan. Please refer to line 107-138 from page 6 to page 8.

In the section of Methods, we have added the theoretical description of how each fear-arousing element makes people afraid. Please refer to line 299-309 of page 17.

Comment #2: 

The authors make an assumption that mask usage (or searching) can act as an indicator that the public is taking steps to protect themselves. I understand that is a reasonable assumption in Taiwan. It might be worth discussing the variation that exists on this dimension cross-nationally. We wouldn’t be able to use mask searching or usage as such an indicator when public health authorities actively dismissed their usage in many other countries until several months into the pandemic (see Merkley & Loewen, 2020).

Response #2:

We are very thankful to the reviewer for this helpful and constructive suggestion. We have added some descriptions about the potential national or cultural differences in the attitudes toward the use of face masks during the COVID-19 epidemic. Also, we remind readers that it should be careful when interpreting Google search volume for fact maks as public epidemic awareness n the context of cross-country comparison. Please refer to line 441-452 of page 26.

Moreover, in the section of conclusion, we have added several sentences to acknowledged that using searching behavior of face masks as an indicator of public epidemic awareness cannot be used in all countries. However, the purpose of the study did not advocate using a universal indicator for measuring or monitoring public epidemic awareness. Instead, we argued that finding out an appropriate digital footprint of public epidemic awareness in a given scenario for a region or a country is necessary since it provides real-time information on public epidemic awareness. Please refer to line 696-702 of page 43.

Comment #3: 

Which brings me to a much more substantive point: government communications are notably absent from this analysis. As noted, I’m not entirely clear on the argument advanced here. But it seems like there is an assumption that cases generated media and public awareness, and subsequently searching related to masks. Of course, cases increased in other Asian countries, but not the same level of searching for masks, which could be in large part due to the seriousness and promptness with which their governments acted on the pandemic and the mitigation strategies they communicated to their mass publics. I think there would be a lot of value added to the analysis if there was some effort to identify a link between government communication and media/the public, along the lines of what we see in agenda setting literature (see Soroka 2002 and many others along those lines)

Response #3:

We are very thankful to the reviewer for this helpful and constructive suggestion. First of all, the study aims to obtain valuable experience based on Taiwan’s response to COVID-19, involving citizens, news media, and the government. Indeed, there is an assumption that confirmed cases of COVID-19 in Taiwan generated media and public awareness, and subsequently searching related to masks. Our results also suggested that news media and the public immediately attached great importance to the imported cases of COVID-19 in Taiwan. 

In the revised manuscript, we have added some discussion about the role of government communication. We emphasize that despite the matter of media on the spread of COVID-19 information, the role of government communication cannot be neglected when it comes to the rise of public epidemic awareness. We discussed how government communication in a transparent, accurate, and timely manner on epidemic control in Taiwan via holding daily press briefing. Please refer to line 574-593 from page 36 to page 37.

Moreover, in the original manuscript of the results section, we put much effort into a discussion about how government measures and public epidemic awareness mutually affect each other, which provides another perspective on the government-public relationships. Please refer to line 460-514 from page 27 to page 29.

Comment #4: 

My biggest concerns relate to the data and methods. First, much more information needs to be given about the precise nature of the data being used in this manuscript, especially related to social media. We need more information on what data KEYPO collects and from what platforms. “Social media” is a very broad term. What exactly is the structure of the data that is provided? How is it accessed? What keywords are used? What are its limitations? None of this is transparently discussed in the manuscript.

Response #4:

We are very grateful to the reviewer for this helpful and constructive suggestion. Also, we are sorry that the information about the data and method in the previous version is not enough for readers to understand the research method of the study. In the revised version, we have uploaded supporting information called “Appendix S1. Information about the scope of the data and KEYPO big data analytics engine.” 

In Appendix S1, we introduced the research subjects, data source, the time range of data, keywords for analysis, and techniques for retrieving data in the study. We hope this information is helpful for the reviewer to understand the methods used in the study. Please refer to Appendix S1. 

Comment #5: 

- A minor, but related point. Wikipedia is not an acceptable source for students, and is not an acceptable source for published research. These data need to be verified from the original source.

Response #5:

Thank you for the suggestion. Regarding the global number of confirmed cases between December 31 and January 19, we have replaced the source of Wikipedia with WHO released data (https://covid19.who.int/). Please refer to line 230-231 of page 13.

The results associating with the global number of confirmed cases have also been modified, including Figures 1 and 2. 

Comment #6: 

Second, I do not think the authors are taking the time series nature of their data seriously enough. There is temporal dependence between these data that do not lend themselves well to simple correlations. For example, confirmed cases will be mechanically related to prior values (in fact, these variables are unit root by definition), while others, like social media and news reports may be (trend) stationary. Autocorrelation

Response #6:

We are very grateful to the reviewer for this helpful and constructive suggestion. In the revised version of the manuscript, we have changed the way we analyze the relationships among time series variables, including the number of confirmed cases of COVID-19, volumes of mentions of COVID-19 and face masks, and Google search volume for COVID-19 and face masks. We adopted time series analysis (TSA) conducted with SPSS version 22 to deal with the nature of time-series data. Specifically, three functions, namely, the autocorrelation function (ACF), the partial autocorrelation function (PACF), and the cross-correlation function (CCF) were implemented. The first two functions, ACF and PACF, were used to identify the temporal dynamics of an individual time series variable and help researchers to examine whether the errors of a certain series variable would correlate itself over time. If a certain variable has autocorrelative relationships, the series data was whitened by removing the autocorrelation before conducting the CCF. The CCF was used to examine the dynamic relationship between two time series variables by calculating the correlation coefficients between two time series variables contemporaneously and at various lagged values. It gives us an insight into how two time series variables coincidently related to each other, and whether movement in one variable tends to precede or follow movement in the other. Please refer to line 264-278 of page 15. The corresponding results of time series analysis were also reported. Please refer to line 336-354 from page 19 to page 21 and line 384-394 of page 23.

---

## [Decision Letter · Decision Letter 2]

18 Mar 2021

PONE-D-20-17953R2

Evolving public behavior and attitudes towards COVID-19 and face masks in Taiwan: A social media study

PLOS ONE

Dear Dr. Chin,

Thank you for submitting your manuscript to PLOS ONE. After careful consideration, we feel that it has merit but does not fully meet PLOS ONE’s publication criteria as it currently stands. Therefore, we invite you to submit a revised version of the manuscript that addresses the points raised during the review process.

The reviewers find a significant improvement was made in this revision, but still point out lingering issues. Particularly, the Introduction needs to clarify how this study builds on past hypotheses. Please address the reviewers’ remaining concerns.

We look forward to receiving your revised manuscript.

Kind regards,

Chang Sup Park, Ph.D.

Academic Editor

PLOS ONE

Journal Requirements:

Reviewers' comments:

Reviewer's Responses to Questions

**Comments to the Author**

1. If the authors have adequately addressed your comments raised in a previous round of review and you feel that this manuscript is now acceptable for publication, you may indicate that here to bypass the “Comments to the Author” section, enter your conflict of interest statement in the “Confidential to Editor” section, and submit your "Accept" recommendation.

Reviewer #3: (No Response)

Reviewer #4: All comments have been addressed

2. Is the manuscript technically sound, and do the data support the conclusions?

Reviewer #3: Partly

Reviewer #4: Yes

3. Has the statistical analysis been performed appropriately and rigorously? 

Reviewer #3: I Don't Know

Reviewer #4: I Don't Know

4. Have the authors made all data underlying the findings in their manuscript fully available?

Reviewer #3: Yes

Reviewer #4: Yes

5. Is the manuscript presented in an intelligible fashion and written in standard English?

Reviewer #3: Yes

Reviewer #4: Yes

6. Review Comments to the Author

Reviewer #3: General comment

This paper is quite innovative in its focus on the use of the social media to measure public reaction to epidemics, and a good resource material for governments and health workers. However, the near absence of organized progression of thoughts in the introduction makes it hard for a reader to understand the hypotheses the authors are testing.

1) I would like to suggest editing the introduction section. What the authors intend to achieve with this paper could be better aligned; clearly and succinctly reflected at the relevant place in the manuscript – at the end of the introduction.

2) The authors need to clarify whether this article is building on past hypotheses or examining the role of the media and government in increasing public epidemic awareness or both. The above is not clear from the introduction.

3) The aims of the study could be carefully pulled together and distinct from the Introduction in lines 102-105, 132 - 137, 166 -172 and 182 -184. And the statement in 181-184 on content analysis should be reflected as an objective of the study if that was the intention.

Please refer to aims of the study mentioned in different sections. These are listed below. I think they can be pulled together/aligned.

(i) Lines 102 – 105. “Given that, one of the purposes of the study is to examine the dynamic relationships between epidemic development and the disease related information spread on the Internet, and to explain how it contributes to the epidemic control during the early period of the COVID-19 epidemic in Taiwan.”

(ii) Lines 132 -137 -Thus, this study aims to fill the gap in the literature on the diffusion of various fear-related messages on the Internet, including mistrust, severity, loss of control, uncertainty, and susceptibility, drawing from previous literature on fear and risk perception [14, 21-23], and their effects on public epidemic awareness along with the development of the COVID-19 epidemic.

(iii) Lines …166 -172………….. This naturalistic study aimed to capture public epidemic awareness of COVID-19 through collecting social media- and Internet-based data, and attempted to elaborate on how the public epidemic awareness rose, and how it played a role in contributing to the successful epidemic prevention in Taiwan during the spread of COVID-19.

(iv) Lines 181 – 184 … Moreover, this study attempted to analyze the transmission of different types of fear information of COVID-19 on the Internet from December 31, 2019 to March 29, 2020, and their effects on the public epidemic awareness by conducting a content analysis.

4) There are doubtful assumptions in this manuscript.

(i) Awareness will most likely lead to adoption of appropriate preventive measures

(ii) The more the media coverage and social media comments the more likely a reduction of prevalence will be achieved due to increased adoption of preventive measures by the public – without taking into consideration the contents of the social media/mass media

(iii) Fear predisposes positive action.

(iv) Increase in case incidence (or rather the announcement/reporting of it) during a pandemic/epidemic will lead to greater media attention and social media activity on that epidemic. It may, on a very short period. Incidentally, the authors indicated that government actions such as the banning of export of facemasks rather than the number of new cases led to rise/decrease in social media activities/mentions. These may have as well influenced adoption of preventive behaviours by the public.

Some of these assumptions have been acknowledged by the authors but the evidence put forward in this paper do not support most of the assumptions. It is important the authors rethink these and provide additional evidence.

Specific Comments

Lines 22 – 23: “This study aims to capture public epidemic awareness of COVID-19 through collecting social media” – ‘document’ maybe a better word to use rather than ‘capture’.

Lines 24 – 26: … “and elaborating on how public epidemic awareness rose and played a role in epidemic prevention in Taiwan during the initial phase of the spread of COVID-19”. This needs to be rephrased. How "awareness" played a role in epidemic prevention (i.e. risk mitigation) is not sufficiently documented in the paper.

Lines 40 – 41: “Moreover, online social media promptly reflect public epidemic awareness, which can be used as a reference for epidemic prevention”. This should be rephrased to indicate the social media activities could be a reflection of public epidemic awareness which can be harnessed for epidemic control.

Lines 58 – 60: “Combining various viewpoints to look at the effectiveness of COVID-19 prevention in Taiwan might be instructive for other countries in dealing with the next crisis resulting from emerging infectious diseases.” This conclusion is not based by the data. Perhaps, the essence of this paper should be on assessing information-seeking by the public during the COVID-19 pandemic.

Lines 67 – 69: ‘However, COVID-19 is more contagious than severe acute respiratory syndrome (SARS) and is infectious even through asymptomatic carriers’ – while this is not in doubt one wonders why this comparison is coming up without previous reference to SARS in this paragraph. A simple sentence on the fact that COVID-19 is very contagious will be helpful.

Lines 72 – 73: ‘Public epidemic awareness has been recognized as a crucial determinant of public epidemic prevention’ – this needs to be supported by literature

Lines 76 – 79: ‘Recent research on public health awareness during the COVID-19 epidemic also showed that one’s awareness of how to prevent COVID-19 significantly contributes to one’s behavioral chance of fighting against COVID-19 [10]’. Only one article is cited to support this statement. I recommend the statement be modified to reflect this e.g. citing the publication specifically.

Lines 166 – 188: These lines can better be summarized and synchronized with information in other parts as to what the study intends to achieve.

Lines 218 -225. Justify the selection of two out of seven indicators (volume of mentions of COVID 19 and the volume of mentions of face masks) as indicators of public epidemic awareness.

Lines 301 – 302: Some repetition - edit

Lines 461 -500. This is a well-articulated section.

3) Lines 488 -491. However, the concluding statement in lines 488-491 “Such positive actions reduced public anxiety about not being able to buy face masks; accordingly, the volume of the mentions and search query of face masks dropped significantly and became less affected by new cases of infection” needs a reference or a supporting data of this study to justify the claim that a “search query of face masks dropped significantly and became less affected by new cases of infection”.

4) Lines 497-500. I am not convinced the data of this study justifies the authors conclusion that masks prevent epidemic of fear —see authors concluding statement “…masks, thus preventing an epidemic of fear.”

Lines 509 – 510: ‘It should be noted that keeping the public epidemic awareness at a high level might be crucial for combating COVID-19.’ This, in my opinion, is the best conclusion of this study.

Lines 516 -517. This is unclear, please explain “According to the ANOVA results (Table 2), the types of fear had a significant effect on the Google search volume of face masks [F(3,358) = 5.67, p < .05].”

Lines 554 – 555: I think this is an overgeneralization. There are other factors that contribute.

Lines 693 – 695 essentially the same as lines 703 – 705. I suggest to edit

Reviewer #4: The authors have done a fine job in revising and enhancing their manuscript and completing their shortcomings. I enjoyed reviewing the manuscript. My only comment is that although the authors' data doesn't cover practice among the general population, I was wondering if it was possible if they could add a paragraph in the discussion section based on literature regarding the practice towards face masks during their study period and whether it was inline with their evidence?

7. PLOS authors have the option to publish the peer review history of their article (what does this mean?). If published, this will include your full peer review and any attached files.

Reviewer #3: No

Reviewer #4: **Yes: **Reza Shahriarirad

---

## [Author Response · Author response to Decision Letter 2]

2 Apr 2021

Response letter for revised Manuscript PONE-D-20-17953R2 

“Evolving public behavior and attitudes towards COVID-19 and face masks in Taiwan: A social media study”

Authors: Chih-Yu Chin, Chang-Pan Liu, Cheng-Lung Wang

We are sincerely grateful to the reviewers for showing the approval of our manuscript and providing constructive comments. We believe that these comments have helped us enhance the quality of the manuscript. We also have done our best to revise as well as improve the paper according to the comments. Please see our responses to each comment as follows.

Academic editor’s comments

General comments (Academic editor): Please review your reference list to ensure that it is complete and correct. If you have cited papers that have been retracted, please include the rationale for doing so in the manuscript text, or remove these references and replace them with relevant current references.

Response #1:

We are very thankful to the editor for reminding us of checking the correctness of references. We have found that some references (news articles) need permission to browse because they are more than six months away from publication time. We have replaced them. Please see the revised references list below.

New reference lists.

[43] Everington K. No need for healthy people to constantly wear masks: Taiwan CECC. Taiwan News. 2020 Jan 31 [Cited 2020 Mar 10]. Available from: https://www.taiwannews.com.tw/en/news/3868005.

[47] Everington K. Taiwan platform includes over 100 apps showing mask availability in stores. Taiwan News. 2020 Feb 27 [Cited 2020 Mar 8]. Available from: https://www.taiwannews.com.tw/en/news/3882111.

[48] Everington K. Wuhan Virus: Taiwan's ban on face mask exports to last until May. Taiwan News. 2020 February 13 [Cited 2020 Mar 8]. Available from: https://www.taiwannews.com.tw/en/news/3875745.

Reviewer 3’s comments

General comment #1 (Reviewer 3): 

(1) I would like to suggest editing the introduction section. What the authors intend to achieve with this paper could be better aligned; clearly and succinctly reflected at the relevant place in the manuscript – at the end of the introduction.

3) The aims of the study could be carefully pulled together and distinct from the Introduction in lines 102-105, 132 - 137, 166 -172 and 182 -184. And the statement in 181-184 on content analysis should be reflected as an objective of the study if that was the intention.

Please refer to aims of the study mentioned in different sections. These are listed below. I think they can be pulled together/aligned.

(i) Lines 102 – 105. “Given that, one of the purposes of the study is to examine the dynamic relationships between epidemic development and the disease related information spread on the Internet, and to explain how it contributes to the epidemic control during the early period of the COVID-19 epidemic in Taiwan.”

(ii) Lines 132 -137 -Thus, this study aims to fill the gap in the literature on the diffusion of various fear-related messages on the Internet, including mistrust, severity, loss of control, uncertainty, and susceptibility, drawing from previous literature on fear and risk perception [14, 21-23], and their effects on public epidemic awareness along with the development of the COVID-19 epidemic.

(iii) Lines …166 -172………….. This naturalistic study aimed to capture public epidemic awareness of COVID-19 through collecting social media- and Internet-based data, and attempted to elaborate on how the public epidemic awareness rose, and how it played a role in contributing to the successful epidemic prevention in Taiwan during the spread of COVID-19.

(iv) Lines 181 – 184 … Moreover, this study attempted to analyze the transmission of different types of fear information of COVID-19 on the Internet from December 31, 2019 to March 29, 2020, and their effects on the public epidemic awareness by conducting a content analysis.

Response #1:

We are very thankful to the reviewer for this helpful and constructive suggestion. Regarding the general comments (1) and (3), in this revised version of the paper, we have combined the research purposes, scattered initially in different paragraphs, at the end of the introduction and make our contentions more understandable for readers. 

At the end of the Introduction section, we split the research purposes into. The first is to document real-time public epidemic awareness of COVID-19 through collecting disease-related information-seeking behaviors on Google in Taiwan and examines its dynamic relations to epidemic development, disease-related information spread on the Internet and government face-mask related policies during the initial phase of the spread of COVID-19 in Taiwan. The other is to investigate the diffusion of various fear-related messages on the Internet and their effects on public epidemic awareness along with the development of the COVID-19 epidemic. Please refer to line 173-194.

General comment #2 (Reviewer 3): 

2) The authors need to clarify whether this article is building on past hypotheses or examining the role of the media and government in increasing public epidemic awareness or both. The above is not clear from the introduction.

Response #2:

We are very grateful to the reviewer for the reminders of unclear statements. Since the study, building on previous research emphasizing the effectiveness of earlier public epidemic awareness on epidemic prevention provides another story about how public epidemic awareness rose and how it was associated with epidemic development, media, and government in Taiwan during the development of the COVID-19 epidemic. Therefore, in this reversion, we add and revise some paragraphs to focus on the government’s and media’s role in arousing the public epidemic awareness. 

The revised paragraphs are listed below.

Lines 67-72: “The study aims to fill in the story of Taiwan’s experience of defending against COVID-19 at an early stage from the perspectives of public epidemic awareness and the prominent role of media in the transmission of COVID-19-related information. Also, the study documents the government’s face-masks countermeasures against COVID-19 and how it associates with public epidemic awareness.”

Lines 85-93: “During a public health crisis, such as emerging infectious disease, effective health communication is vital for the rise of public epidemic awareness, where government and media play crucial roles [11, 12, 13, 14]. As for a government, it is essential to adopt an effective communication strategy with useful content, trustworthy sources, and efficient channels to enable the public to be aware of or learn about disease-related information in a timely manner. Previous research has indicated the higher public’s exposure to government information about COVID-19, the public would have higher probabilities to take necessary preventive measures to protect themselves from being infected by COVID-19 [11].”

References 11 and 12 are the newly added articles regarding the role of government communication:

[11] Chang C. Cross-country comparison of effects of early government communication on personal empowerment during the COVID-19 pandemic in Taiwan and the United States. Health Commun. https://doi.org/10.1080/10410236.2020.1852698

[12] Freimuth V, Linnan HW. Potter P. Communicating the threat of emerging infections to the public. Emerging Infect. Dis.2000; 6(4): 337–347.

General comment #3 (Reviewer 3): 

There are doubtful assumptions in this manuscript.

(i) Awareness will most likely lead to adoption of appropriate preventive measures

(ii) The more the media coverage and social media comments the more likely a reduction of prevalence will be achieved due to increased adoption of preventive measures by the public – without taking into consideration the contents of the social media/mass media

(iii) Fear predisposes positive action.

(iv) Increase in case incidence (or rather the announcement/reporting of it) during a pandemic/epidemic will lead to greater media attention and social media activity on that epidemic. It may, on a very short period. Incidentally, the authors indicated that government actions such as the banning of export of facemasks rather than the number of new cases led to rise/decrease in social media activities/mentions. These may have as well influenced adoption of preventive behaviours by the public.

Some of these assumptions have been acknowledged by the authors but the evidence put forward in this paper do not support most of the assumptions. It is important the authors rethink these and provide additional evidence.

Response #3:

We are very thankful to the reviewer for this helpful and constructive suggestion. As the reviewer mentioned, many of our assumptions are based on previous literature rather than our provided evidence. Therefore, in this revised version, we have changed the statements, especially the sub-title, and focused on describing how Taiwan's public epidemic awareness emerged and the media and the government's role in it, trying to remove these doubtful assumptions. 

Regarding the first doubtful assumption that awareness will most likely lead to adoption of appropriate preventive measures, we have substituted the too strong statement entitled “Rapidly increasing public epidemic awareness effectively prevent its spread” with “The role of news media and government in rapidly increasing public epidemic awareness in the early stage of the spread of COVID-19.” Please refer to lines 173-194. We also reviewed the entire content under the sub-section. We emphasize the rise of public epidemic awareness and its relations to the news media and the government in the revised version, It seems that the content is more in line with the revised titile. 

As for the second doubt assumption, we are sorry that we are wondering whether we made this hypothesis. We mentioned in the introduction that when emerging infectious diseases are heavily and quickly discussed in media, the public can better prepare and prevent them earlier (in lines 99-105). However, in our results or the discussion section, we did not mention that a reduction of prevalence in Taiwan is because the people have been affected by the mass media or social media discussing COVID-19 and then have taken prevention measures. What we emphasize is the part of the rise of public epidemic awareness, so we add a sentence into the paragraph. Please refer to line 589.Moreover, , we added a paragraph about the limitation and future suggestions at the end of the discussion as below: 

“Finally, our research used the web methodology to collect timely information about public epidemic awareness and the quantity of disease-information spread on the Internet and examine the dynamic relationships. However, we did not consider the content of information transmitted on the Internet. Future studies might adopt various analytics such as text mining and sentiment analysis to take what the news spread into account. Please refer to line 712-717.”

Regarding the third assumption that fear predisposes positive action, we have replaced the statement entitled “The role of fear in stimulating epidemic awareness” with “Types of fear-embedded popular events and public epidemic awareness in the course of COVID-19 spread in Taiwan.” Please refer to lines 672-673. We removed the sentences that comparing the loss of control is more substantial than perceived susceptibility and perceived severity to cause people to wear a face mask. Moreover, we have pointed out more euphemistically: the loss of control might explain why panic behavior and massive searching behaviors for face masks occur in Taiwan (Please refer to line 696-699). 

As for the fourth assumption, we have removed the statement that “the volume of the mentions and search query of face masks became less affected by new cases of infection.” Please refer to lines 500-503. 

Specific comment #4 (Reviewer 3): 

Lines 22 – 23: “This study aims to capture public epidemic awareness of COVID-19 through collecting social media” – ‘document’ maybe a better word to use rather than ‘capture”.

Response #4:

Thank you for choosing a better word in this context. We have replaced the word “capture” with “document.” Please refer to line 23 and 172.

Specific comment #5 (Reviewer 3): 

Lines 24 – 26: … “and elaborating on how public epidemic awareness rose and played a role in epidemic prevention in Taiwan during the initial phase of the spread of COVID-19”. This needs to be rephrased. How "awareness" played a role in epidemic prevention (i.e. risk mitigation) is not sufficiently documented in the paper.

Response #5:

We are very thankful to the reviewer for this helpful and constructive suggestion. We have modified the purpose of this study to make it more representative of our research. The revised sentences are following:

“This study aims to document public epidemic awareness of COVID-19 in Taiwan through collecting social media- and Internet-based data, and provide valuable experience of Taiwan’s response to COVID-19, involving citizens, news media, and the government, to aid the public in overcoming COVID-19, or infectious diseases that may emerge in the future.” Please refer to lines 24-27.

Specific comment #6 (Reviewer 3): 

Lines 40 – 41: “Moreover, online social media promptly reflect public epidemic awareness, which can be used as a reference for epidemic prevention”. This should be rephrased to indicate the social media activities could be a reflection of public epidemic awareness which can be harnessed for epidemic control.

Response #6:

We are very thankful to the reviewer for this helpful and constructive suggestion. We have rephrased the sentence as below:

“Moreover, in the era of digitalization, online and social media activities could reflect public epidemic awareness which can e harnessed for epidemic control.” Please refer to lines 41-42.

Specific comment #7 (Reviewer 3): 

Lines 58 – 60: “Combining various viewpoints to look at the effectiveness of COVID-19 prevention in Taiwan might be instructive for other countries in dealing with the next crisis resulting from emerging infectious diseases.” This conclusion is not based by the data. Perhaps, the essence of this paper should be on assessing information-seeking by the public during the COVID-19 pandemic.

Response #7:

We are very thankful to the reviewer for this helpful and constructive suggestion. We have rewritten the sentences to a better representative of our research as follows:

“The study aims to fill in the story of Taiwan’s experience of defending against COVID-19 at an early stage from the perspectives of public epidemic awareness measured by disease-related information-seeking by the public and the prominent role of media in the transmission of COVID-19-related information. Also, the study documents the government’s face-masks countermeasures against COVID-19 and how it associates with public epidemic awareness. This might be instructive for other countries in dealing with the next crisis resulting from emerging infectious diseases.” Please refer to lines 67-74.

Specific comment #8 (Reviewer 3): 

 Lines 67 – 69: ‘However, COVID-19 is more contagious than severe acute respiratory syndrome (SARS) and is infectious even through asymptomatic carriers’ – while this is not in doubt one wonders why this comparison is coming up without previous reference to SARS in this paragraph. A simple sentence on the fact that COVID-19 is very contagious will be helpful.

Response #8:

Thank you for suggesting a better way to express the core idea. We have replaced it with a simple sentence as the reviewer suggested as follows:

“However, COVID-19 is very contagious and is infectious even through asymptomatic carriers [7]” Please refer to lines 64-65.

Specific comment #9 (Reviewer 3): 

 Lines 72 – 73: ‘Public epidemic awareness has been recognized as a crucial determinant of public epidemic prevention’ – this needs to be supported by literature

Response #9:

Thank you for reminding us to add some references. Please refer to line 76. 

Specific comment #10 (Reviewer 3): 

 Lines 76 – 79: ‘Recent research on public health awareness during the COVID-19 epidemic also showed that one’s awareness of how to prevent COVID-19 significantly contributes to one’s behavioral chance of fighting against COVID-19 [10]’. Only one article is cited to support this statement. I recommend the statement be modified to reflect this e.g. citing the publication specifically.

Response #10:

Thank you for the reminder. We have added “e.g.,” before the reference [10]. 

Specific comment #11 (Reviewer 3): 

Lines 166 – 188: These lines can better be summarized and synchronized with information in other parts as to what the study intends to achieve.

Response #11:

We are very thankful to the reviewer for this helpful and constructive suggestion. We have combined the purpose of this study into the end of the introduction to make better aligned. Please refer to lines 173-194.

Specific comment #12 (Reviewer 3): 

Lines 218 -225. Justify the selection of two out of seven indicators (volume of mentions of COVID 19 and the volume of mentions of face masks) as indicators of public epidemic awareness. Lines 301 – 302: Some repetition - edit

Response #12:

We are very thankful to the reviewer for this helpful and constructive suggestion. In this reversion, we added a paragraph in the Method section to justify the Google search volume for COVID-19 and face masks as indicators of public epidemic awareness. 

First of all, disease information-seeking behaviors have been used as an indicator of public epidemic awareness. Moreover, we argued that increasing intended behaviors of searching for face masks indicates that the public is aware of the threat and reflects other positive hygiene practices. These are the reasons why we choose the two variables to reflect public epidemic awareness. The paragraph shows below: 

“The disease information searching behaviors have been used as an indicator of public epidemic awareness recently [29]. It reflects the behavioral intentions of individuals. In addition to searching for disease, given that related treatment drugs and vaccines for COVID-19 were under clinical trials in the early stage of COVID-19 [33], face masks have become necessary personal protective equipment for managing infectious diseases [34,35]. Increasing intended behaviors of searching for face masks indicates that the public is aware of the threat and reflects other positive hygiene practices [36,37]. Thus, we examine the public epidemic awareness by measuring the public online information-seeking behavior of COVID-19 and face masks.” Please refer to lines 231-240.

Specific comment #13 (Reviewer 3): 

Lines 301 – 302: Some repetition - edit

Response #13:

Think you for the kindly reminder. The repetition has been edited. Please refer to line 316.

Specific comment #14 (Reviewer 3): 

Lines 488 -491. However, the concluding statement in lines 488-491 “Such positive actions reduced public anxiety about not being able to buy face masks; accordingly, the volume of the mentions and search query of face masks dropped significantly and became less affected by new cases of infection” needs a reference or a supporting data of this study to justify the claim that a “search query of face masks dropped significantly and became less affected by new cases of infection”

Response #14:

We are very thankful to the reviewer for this helpful and constructive suggestion. Since our evidence indeed cannot prove that “the volume of the mentions and search query of face masks became less affected by new cases of infection,” we have removed the statement. Please refer to lines 500-503. 

Specific comment #15 (Reviewer 3): 

Lines 497-500. I am not convinced the data of this study justifies the authors conclusion that masks prevent epidemic of fear —see authors concluding statement “…masks, thus preventing an epidemic of fear.”

Response #15:

We are very thankful to the reviewer for this helpful and constructive suggestion. After we reviewed the sentence, we removed the sentence “this preventing an epidemic of fear.” We agree with the reviewer that it is explained to an excessive degree. Please refer to line 512. 

Specific comment #16 (Reviewer 3): 

Lines 509 – 510: ‘It should be noted that keeping the public epidemic awareness at a high level might be crucial for combating COVID-19.’ This, in my opinion, is the best conclusion of this study.

Response #16:

Thank you for appreciating our argument.

Specific comment #17 (Reviewer 3): 

Lines 516 -517. This is unclear, please explain “According to the ANOVA results (Table 2), the types of fear had a significant effect on the Google search volume of face masks [F(3,358) = 5.67, p < .05].”

Response #17:

Thank you for the reminder of the unclear statement. We have explained the results that the Google search volume for face masks varied by the dissemination of various types of fear-embedded messages. Please refer to lines 530-531.

Specific comment #18(Reviewer 3): 

Lines 554 – 555: I think this is an overgeneralization. There are other factors that contribute.”

Response #18:

We are very thankful to the reviewer for this helpful and constructive suggestion. After careful consideration, we agree with the reviewer's opinion that it is an overgeneralization. Thus, we have substituted the too strong statement entitled “Rapidly increasing public epidemic awareness effectively prevent its spread” with “The role of news media and government in rapidly increasing public epidemic awareness in the early stage of the spread of COVID-19.” Please refer to lines 173-194. 

Specific comment #18(Reviewer 3): 

Lines 693 – 695 essentially the same as lines 703 – 705. I suggest to edit

Response #18:

Think you for kindly reminder. The repetition has been edited. Please refer to lines 727-729.

Reviewer 4’s comments

General comment #1: 

The authors have done a fine job in revising and enhancing their manuscript and completing their shortcomings. I enjoyed reviewing the manuscript. My only comment is that although the authors' data doesn't cover practice among the general population, I was wondering if it was possible if they could add a paragraph in the discussion section based on literature regarding the practice towards face masks during their study period and whether it was inline with their evidence?

Response #1:

Thank you for appreciating our effort in revising our previous work. We are also grateful for those constructive comments which definitely improve the quality of our manuscript.

Regarding the suggestion of adding a paragraph about the practice towards face masks during their study period and whether it was inline with their evidence, we have added some references to show the effectiveness of using face makes and how Taiwan strictly enforce the policy of wearing masks in Taiwan. The added paragraph shows below:

“While our data could not directly reflect one’s actual preventive behavior of wearing a face mask and prove its effectiveness of epidemic prevention, previous literature has documented that Taiwan adopted widespread measures for the public to wear masks would be the reason for the reduction of COVID-19 transmission and even having good control of the COVID-19 without a mandatory suspension of work and school [61]. For example, TCDC requires people to wear masks in public places, including public transport and hospital, and that will be fined if they fail to comply. The other research also showed that Taiwan, a mask-wearing country, has lower growth rates of COVID-19 cases compared to the non-mask-waring counties in the early stage of the spread of COVID-19 via combining mathematical modeling and existing scientific evidence [44].” Please refer to lines 648-658.

References 61 are the newly added article regarding the effectiveness of using face-masks to prevent COVID-19 spread.

[61] Wei J. Guo S. Long E. Zhang L. Shu B. Guo L. Why does the spread of COVID-19 vary greatly in different countries? Revealing the efficacy of face masks in epidemic prevention. Epidemiol. Infect. 2021. https://doi.org/10.1017/S0950268821000108

---

## [Decision Letter · Decision Letter 3]

5 May 2021

Evolving public behavior and attitudes towards COVID-19 and face masks in Taiwan: A social media study

PONE-D-20-17953R3

Dear Dr. Chin,

We’re pleased to inform you that your manuscript has been judged scientifically suitable for publication and will be formally accepted for publication once it meets all outstanding technical requirements.

Kind regards,

Chang Sup Park, Ph.D.

Academic Editor

PLOS ONE

Additional Editor Comments (optional):

Reviewers' comments:

Reviewer's Responses to Questions

**Comments to the Author**

1. If the authors have adequately addressed your comments raised in a previous round of review and you feel that this manuscript is now acceptable for publication, you may indicate that here to bypass the “Comments to the Author” section, enter your conflict of interest statement in the “Confidential to Editor” section, and submit your "Accept" recommendation.

Reviewer #4: All comments have been addressed

2. Is the manuscript technically sound, and do the data support the conclusions?

Reviewer #4: Yes

3. Has the statistical analysis been performed appropriately and rigorously? 

Reviewer #4: Yes

4. Have the authors made all data underlying the findings in their manuscript fully available?

Reviewer #4: Yes

5. Is the manuscript presented in an intelligible fashion and written in standard English?

Reviewer #4: Yes

6. Review Comments to the Author

Reviewer #4: (No Response)

7. PLOS authors have the option to publish the peer review history of their article (what does this mean?). If published, this will include your full peer review and any attached files.

Reviewer #4: **Yes: **Reza Shahriarirad

---

## [Editor Report · Acceptance letter]

10 May 2021

PONE-D-20-17953R3 

Evolving public behavior and attitudes towards COVID-19 and face masks in Taiwan: A social media study 

Dear Dr. Chin:

I'm pleased to inform you that your manuscript has been deemed suitable for publication in PLOS ONE. Congratulations! Your manuscript is now with our production department. 

Kind regards, 

on behalf of

Dr. Chang Sup Park 

Academic Editor

PLOS ONE